# Electrically driven reprogrammable phase-change metasurface reaching 80% efficiency

Sajjad Abdollahramezani[1], Omid Hemmatyar[1], Mohammad Taghinejad[1], Hossein Taghinejad[1], Alex Krasnok [2,10], Ali A. Eftekhar[1], Christian Teichrib[3], Sanchit Deshmukh [4], Mostafa A. El-Sayed[5], Eric Pop [4,6,7], Matthias Wuttig [3], Andrea Alù [2,8], Wenshan Cai [1,9] & Ali Adibi [1✉]

Phase-change materials (PCMs) offer a compelling platform for active metaoptics, owing to their large index contrast and fast yet stable phase transition attributes. Despite recent advances in phase-change metasurfaces, a fully integrable solution that combines pronounced tuning measures, i.e., efficiency, dynamic range, speed, and power consumption, is still elusive. Here, we demonstrate an in situ electrically driven tunable metasurface by harnessing the full potential of a PCM alloy, $Ge_2Sb_2Te_5$ (GST), to realize non-volatile, reversible, multilevel, fast, and remarkable optical modulation in the near-infrared spectral range. Such a reprogrammable platform presents a record eleven-fold change in the reflectance (absolute reflectance contrast reaching 80%), unprecedented quasi-continuous spectral tuning over 250 nm, and switching speed that can potentially reach a few kHz. Our scalable heterostructure architecture capitalizes on the integration of a robust resistive microheater decoupled from an optically smart metasurface enabling good modal overlap with an ultrathin layer of the largest index contrast PCM to sustain high scattering efficiency even after several reversible phase transitions. We further experimentally demonstrate an electrically reconfigurable phase-change gradient metasurface capable of steering an incident light beam into different diffraction orders. This work represents a critical advance towards the development of fully integrable dynamic metasurfaces and their potential for beamforming applications.

[1] School of Electrical and Computer Engineering, Georgia Institute of Technology, Atlanta, GA 30332, USA. [2] Photonics Initiative, Advanced Science Research Center, City University of New York, New York, NY 10031, USA. [3] Physikalisches Institut IA, RWTH Aachen, 52074 Aachen, Germany. [4] Department of Electrical Engineering, Stanford, CA 94305, USA. [5] Laser Dynamics Laboratory, School of Chemistry and Biochemistry, Georgia Institute of Technology, Atlanta, GA 30332, USA. [6] Department of Materials Science and Engineering, Stanford University, Stanford, CA 94305, USA. [7] Precourt Institute for Energy, Stanford University, Stanford, CA 94305, USA. [8] Physics Program, Graduate Center, City University of New York, New York, NY 10016, USA. [9] School of Materials Science and Engineering, Georgia Institute of Technology, Atlanta, GA 30332, USA. [10]Present address: Department of Electrical and Computer Engineering, Florida International University, Miami, FL 33174, USA. ✉email: ali.adibi@ece.gatech.edu

Optical metasurfaces, planar devices comprising densely arranged arrays of patterned nanostructures, extend most functionalities realized by conventional bulky optical components by imparting arbitrary spatial and spectral transformations on incident light waves[1–3]. To enable post-fabrication tuning of metasurfaces, the incorporation of material platforms with tunable properties such as transparent conductive oxides[4], liquid crystals[5], 2D materials[6], doped semiconductors[7], and elastomeric polymers[8] capitalizing on the conventional electro-optic, electro-mechanic, and thermo-optic effects has been envisioned. Despite impressive progress in the realization of tunable metasurfaces, most existing demonstrations suffer from limitations including relatively weak optical modulation strength (due to low quality-factor ($Q$) nanoantennae), low optical performance (imposed by excessive material losses), low-speed modulation (limited by the intrinsic properties of tunable materials), and/or challenging manufacturing (on account of non-complementary metal oxide semiconductor (CMOS)-friendly fabrication processes). In this regard, new efficient material platforms improving the dynamic range of amplitude and phase modulations, facilitating the pixel-level programming, enhancing the modulation speed, and reducing the static power consumption for the next-generation adaptive functional systems are in great demand.

Phase-change materials (PCMs) with optical properties (e.g., refractive index) that are remarkably modified upon crystallization, attributed to the metavalent bonding in the crystalline phase[9], can mitigate such challenges[10–16]. Amongst existing PCMs, archetypal compound germanium antimony telluride ($Ge_2Sb_2Te_5$ or GST for short) has been vastly exploited in commercial rewritable optical disk storage technology and phase-change electronic memory applications exhibits attractive intrinsic features including non-volatility (long retention time of at least 10 years), ultrafast switching speed (10–100 s of ns), high switching robustness (potentially up to $10^{12}$ cycles), considerable scalability (down to nanometer-scale lengths), low energy transition (down to a few $aJ/nm^3$), compatibility with CMOS processes, and good thermal stability, among others[17–23]. Given the unique optical and electrical properties of GST, recently, significant attention has been paid toward the implementation of reconfigurable metadevices based on these properties[24–29].

To date, the dynamic tuning of phase-change metasurfaces has been entangled with active switching of PCMs between the amorphous and crystalline state using thermal-conduction annealing[26] or laser pulses excitation[30]. While the former limits the device performance to one-way amorphous-to-crystalline conversion, both approaches necessitate the use of bulky external apparatus, i.e., a heating stage or an ultrafast laser. Inspired by the technological developments in mature phase-change random access memories, electrical threshold switching holds the promise to precisely control the states of the PCM in individual building blocks of metasurfaces, i.e., "meta-atoms", through a cross-bar array architecture[31]. However, this scheme places stringent constraints on the targeted optical performance of the phase-change metasurface due to (i) interference of lossy metal wires with free-space light incident on the subwavelength meta-atoms, and (ii) formation of crystallization filamentation as a direct current path through PCM that prevents uniform phase transition of the whole PCM volume in meta-atoms. To meet these challenges, we leverage in situ electrical Joule heating using an optimized microheater design that indirectly actuates the PCM element of our metasurface. Our demonstration largely surpasses the state-of-the-art alternatives for reconfigurable metasurfaces in four important dimensions. First, the heterostructure metadevice platform formed by the integration of a robust microheater underneath the metasurface enables uniform electrothermal phase conversion without adding excessive dissipative loss to the

optical device. Reaching 80% optical efficiency, our platform outperforms the recently developed reflector-absorber PCM-switches[32,33]. In addition, our architecture significantly reduces the deformation of the meta-atoms caused by inevitable heating of the alternative resistive microheaters[33] that use plasmonic materials like silver with low melting temperature. Second, our electrically driven platform enables achieving multiple non-volatile intermediate PCM states (between amorphous and crystalline) in a repeatable fashion to realize multi-state reconfigurable metasurfaces necessary for adaptive flat optics. Third, we use technologically mature GST that provides highest index contrast among all PCMs at the near-infrared (near-IR) spectral range. This allows us to fairly decrease the thickness of the PCM layer without compromising the optical performance. The application of a thin PCM layer is necessary for the realization of a repeatable and reliable amorphization process while avoiding elemental segregation as a typical failure mechanism of PCMs during melt-quenching[19,21]. Furthermore, orders-of-magnitude faster registration of freely accessible intermediate states with lower dynamic power can be performed in GST in comparison to the recently emerged phase-change alloys such as $Ge_2Sb_2Se_4Te_1$ (GSST)[32]. Finally, the fundamental modes of the metasurface on account of the near-field interaction of the incident light with plasmonic elements exhibit good modal overlap with the GST film, which facilitates the manipulation of the optical scattering with a wide-dynamic range.

Here, we demonstrate on-demand optical modulation and wavefront engineering using an electrically driven, fully reversible, and reconfigurable GST-based metasurface with multiple intermediate states and a large tuning range through judicious co-optimization of a multiphysics model taking into account the extreme electrical, thermal, and optical properties of the contributing materials. Our platform has the potential for major applications in several fields including imaging, computing, and sensing.

## Results and discussion

**Electrothermal analysis of the integrated heterostructure metadevice.** A schematic view of the reconfigurable heterostructure metadevice driven by in situ electrical pulses is represented in Fig. 1a(i). The microheater consists of a 12 μm × 12 μm square of 50 nm-thick tungsten (W) layer connected to the top metasurface with a 20 nm-thick layer of alumina ($Al_2O_3$) and isolated from the silicon (Si) substrate, as a good heat sink, with a 100 nm-thick hafnia ($HfO_2$) film. We chose W for the microheater material due to its highest melting point, good thermal conductivity, moderate resistivity, and low thermally activated diffusion. While high thermal conductivity of $Al_2O_3$, in comparison to all existing oxides, significantly facilitates heat exchange between the microheater and the metasurface, a thick-enough $HfO_2$ layer helps preservation of the generated heat for GST phase change to keep the electrical power consumption low. The fabrication processes are detailed in "Methods" and Supplementary Fig. S1.

In order to study the performance of the miniaturized heater in terms of temperature uniformity, switching speed, heating/cooling rate, and operational voltage, the real-time temperature distribution of the heterostructure metadevice in response to two types of electrical pulses with different temporal profiles (i.e., "set" and "reset") are calculated. As shown in Fig. 1b(i), the low-voltage set pulse (with 200 μs-long double exponential waveform and a peak voltage of 1.7 V) heats amorphous GST (A-GST for short) above the crystallization temperature (~160 °C[34]) for a sufficiently long time to ensure full nucleation and formation of monolithic crystalline islands. By decreasing the voltage amplitude, arbitrary

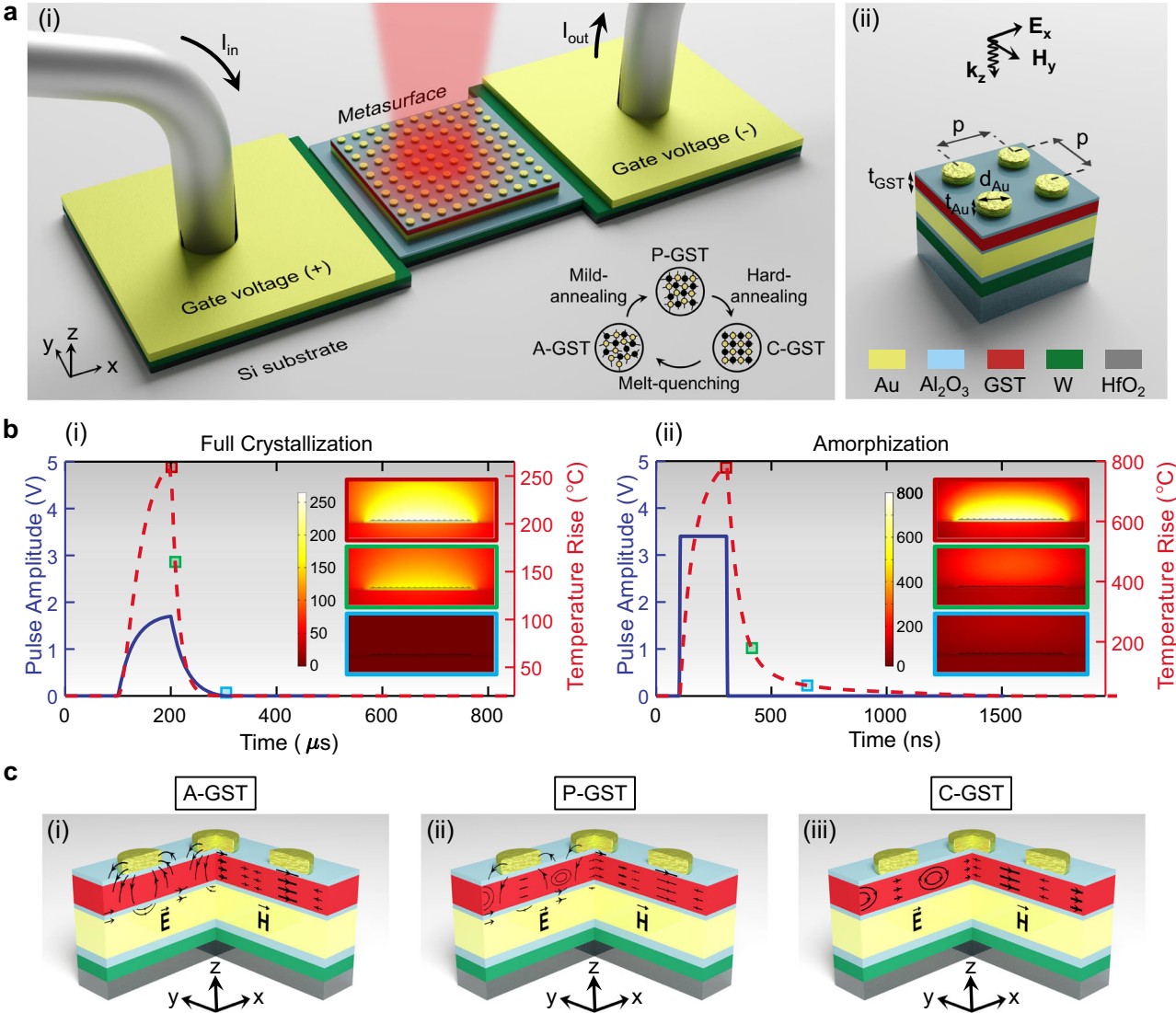

**Fig. 1 Operation principle of an electrically reconfigurable heterostructure metadevice. a** (i) A perspective view of the heterostructure consisting a resistive microheater integrated with a phase-change metasurface. The microheater is constructed by deposition of a resistive patch of W connecting two Au probing pads on top of a HfO$_2$-coated Si substrate. The metasurface is formed by multilayer deposition of an optically-thick Au backreflector, a GST film encapsulated between two equally-thick protective layers of Al$_2$O$_3$, and a 2D array of Au nanodisks. Inset: a generic scheme of atomic distribution of partial crystalline GST (P-GST), fully crystalline GST (C-GST), and amorphous GST (A-GST) after mild-annealing by applying a long low-intensity pulse, hard-annealing by applying a long medium-intensity pulse, and melt-quenching by applying a short high-intensity pulse. (ii) Cross section view of the heterostructure metadevice. **b** Real-time voltage of the applied "set" and "reset" pulses (solid blue lines) and the corresponding temperature responses (dashed red lines) in the center of the GST film for (i) full crystallization and (ii) amorphization processes. Inset: simulated temperature distributions at the cross section of the phase-change metasurface parallel to the current flow at the time marked by the color-coded markers. **c** The electric and magnetic field vector distributions for (i) SR-SPP, (ii) the hybrid mode, and (iii) LR-SPP within the x–z and y–z planes of a meta-atom, respectively.

fractions of amorphous and crystalline states can also be registered in the material that results in partial crystalline GST (P-GST for short). On the other hand, a high-voltage pulse (with 200 ns-long rectangular waveform and a peak voltage of 3.4 V) featuring very short leading/trailing edge (~5 ns) rapidly increases the temperature of P-GST or fully crystalline GST (C-GST for short) above the melting temperature (~630 °C[34]) followed by quenching such that GST solidifies in the amorphous state (see Fig. 1b(ii)). Considering the enlarged size of the microheater (one order of magnitude larger than those previously studied[13,35,36]), uniform heat generation over the whole volume of the GST film should be carefully addressed to ensure reliable and repeatable optical performance. The simulated two-dimensional (2D) temperature map in Fig. 1b(i) indicates that at the end of the

set pulse, the temperature difference between the center and the two ends of the metasurface is <12% while the temperature gradient along the out-of-plane direction is <0.2% (see Supplementary Note 1 and Figs. S2–S4 for more details on the local and global addressing of the GST film, respectively). Another grand challenge is the realization of sufficiently fast cooling rate of PCMs in the amorphization process. To speed up the transient thermal response of the metadevice, an optimized selection of materials and geometries for the microheater, metasurface, and the surrounding medium is considered (see "Methods" and Supplementary Note 1). As depicted by the electrothermal simulations in Fig. 1b(ii), the heterostructure with an elevated temperature of ~790 °C at the end of the reset pulse is cooled down with the rate of ~10 °C/ns to <480 °C and ~6 °C/ns

to <160 °C, which is higher than typical 1 °C/ns melt-quenching criterion[19,21]. These features are essential to precisely register multiple reversible intermediate phases to the GST film and enabling reprogrammable multifunctional metasurfaces, a key attribute of our work (see Supplementary Note 1 and Fig. S2 for more details).

**Design of the optically smart hybrid phase-change metasurface.** The 10.5 μm × 10.5 μm phase-change metasurface comprises 35 nm-thick gold (Au) nanodisks separated from a 80 nm-thick Au backreflector by a 40 nm-thick blanket film of GST sandwiched between two 10 nm-thick layers of $Al_2O_3$ (see Fig. 1a(ii); also, see "Methods" for fabrications details). Two $Al_2O_3$ layers prevent heating-induced oxidation of the GST film and diffusion of the noble metal into GST during the heating process. The hybrid plasmonic-PCM meta-atom can support two distinctive surface plasmon polariton (SPP) modes, namely long-range SPP (LR-SPP) and short-range SPP (SR-SPP) (see "Methods" and Supplementary Fig. S6 for details on the origination and evolution of these modes). The spatial characteristics of the SR-SPP and LR-SPP modes due to the excitation of SPPs at the finite interface of an individual nanodisk and the top $Al_2O_3$ layer and that at the infinite interface of the Au backreflector and the bottom $Al_2O_3$ layer are schematically illustrated in Fig. 1c(i) and c(iii), respectively. The electric and magnetic flowlines in Fig. 1c(ii) reveal the existence of a hybrid mode due to the overlap between the localized SR-SPP and distributed LR-SPP modes. The rich physical properties and distinct characteristics of governing modes coupled to the available state of GST offer a good degree of freedom for realization of multifunctional metasurfaces. Particularly, the evolution of the governing mode from SR-SPP in A-GST (with higher plasmonic and lower photonic loss) to overlapped SR-SPP/LR-SPP in P-GST (with balanced plasmonic and photonic losses) and finally to LR-SPP in C-GST (with lower plasmonic and higher photonic loss) facilitates manipulation of both amplitude and phase properties of light in the telecom wavelengths.

**Dynamically reprogrammable metasurface characterization.** We leverage the rich nature of the SPP modes and large index contrast of GST for the demonstration of an electrically driven multispectral meta-switch. Figure 2a(i) shows the fabricated sample (see Methods for details) mounted on a ceramic chip carrier. A scanning electron microscope (SEM) image of the heterostructure metadevice and a bird's-eye view of the metasurface consisting of an array of 17 × 17 identical meta-atoms are depicted in Fig. 2a(ii) and a(iii), respectively. We use a home-built reflectometry setup coupled to two signal generators for co-located electrical excitation and optical measurement of fabricated samples (see Fig. 2b and "Methods" for details).

The structural design parameters of the metasurface are judiciously chosen to keep fundamental resonances of the meta-switch at the two extreme cases of GST far apart, which guarantees high modulation depth over a wide spectral bandwidth. To corroborate the design strategy, the statistical distribution of experimentally measured reflectance spectra over 50 consecutive cycles of crystallization-amorphization is displayed in Fig. 2c (see Supplementary Fig. S7 for the detailed reflectance spectra of all cycles). The narrow boxes reveal the slight deviation of the first and the third quartiles from the median of sampled data for 19 discrete wavelengths in the telecom range. It is evident that the resonance wavelength of the meta-switch red shifts from 1390 nm with A-GST to 1640 nm with C-GST. Upon this transition, a record average absolute ($\Delta R = |R_{A\text{-GST}} - R_{C\text{-GST}}|$) and relative modulation depth ($\Delta R/R_{C\text{-GST}}$) over 75 and 1000% are achieved at

1640 nm, respectively. More importantly, with average 82% reflectance in the reflective mode, our platform surpasses the state-of-the-art electrically tunable PCM meta-switches[32,33]. For more clarity, time-dependent traces of the change in the reflectance at 1640 nm during consecutive cycles of switching are depicted in Fig. 2d. The measured 95% confidence intervals (shaded areas) of ±1 and ±7.5% for the reflective and absorptive state, respectively, verify the highly reproducible switching process. Such consistent characteristics are also verified through confocal Raman microscopy by studying the micro-Raman scattering from the meta-switch after applying 2 consecutive cycles of set/reset electrical pulses (see Fig. 2e). The non-deterministic behavior of the absorptive and reflective state stems from the formation of non-homogeneous crystalline regions during heating and stochastic recrystallization of small islands in GST during the quenching process, respectively[17,31,34]. Considering the fast relaxation time of GST and small thermal time constant and temperature non-uniformity of the heterostrucutre metadevice (as verified by electrothermal analysis in Fig. 1b(i)), a meta-switch with the operation speed of a few kHz can be realized with order-of-magnitude lower operational voltages compared to the state-of-the-art electro-mechanical optical metasurfaces[37]. Moreover, our platform enables three orders-of-magnitude faster registration of the crystalline state with lower dynamic power in GST in comparison to other platforms using emergent PCMs with slow response time such as GSST[32].

**Active multistate tuning of the phase-change metasurface.** Besides the binary-level switching, distinctive and stable intermediate crystallographic states of GST, in virtue of its giant index contrast, non-volatile, and nucleation-dominant characteristics, hold the promise for multi-state switching operation. Considering the good thermal uniformity across the microheater, precise control of the crystalline fraction of GST, through the formation of critical nuclei and their subsequent growth, can be realized by applying a customized electrical pulse. We explore this capability by programming the meta-switch with fixed length pulses featuring different voltages. The measured reflectance of the meta-switch for A-GST, C-GST, and 4 accessed intermediate phases of GST (see Fig. 3a(i)) show quasi-continuous tuning of the fundamental resonances (i.e., LR-SPP and SR-SPP) from 1390 to 1640 nm. With such record optical contrast, unprecedented ultrawide spectral tuning range, and potential fast switching operation, our platform outperforms many existing works relying on electro-optical, electro-mechanical, and thermo-optic effects[38].

To quantitatively analyze the crystallization kinetics upon electrical pulse excitation, we compare the measured reflectance spectra with simulated ones for different crystallization fractions of GST, whose optical properties are approximated using an effective medium theory (see Supplementary Note 2). As shown in Fig. 3a, a good agreement is observed between the color-coded experimental measurements and simulated results from intermediate states with ~20% crystallization steps. Figure 3b depicts the correlation between the measured applied pulse voltage, resonance wavelength of the meta-switch, and approximated crystallization fraction. Evidently, a pronounced tuning range is achieved upon multi-state conversion of GST using electrical pulses with small voltages. We further quantitatively investigate the crystallization fractions of GST in different intermediate states as a function of the induced temperature (see Supplementary Note 1 and Fig. S2b).

To study the physical mechanism behind the operation of the meta-switch, we calculate the electromagnetic field distribution at the two dips of the reflectance spectrum for the intermediate case with 80% crystallization fraction (see Fig. 3a(ii)). The field profiles

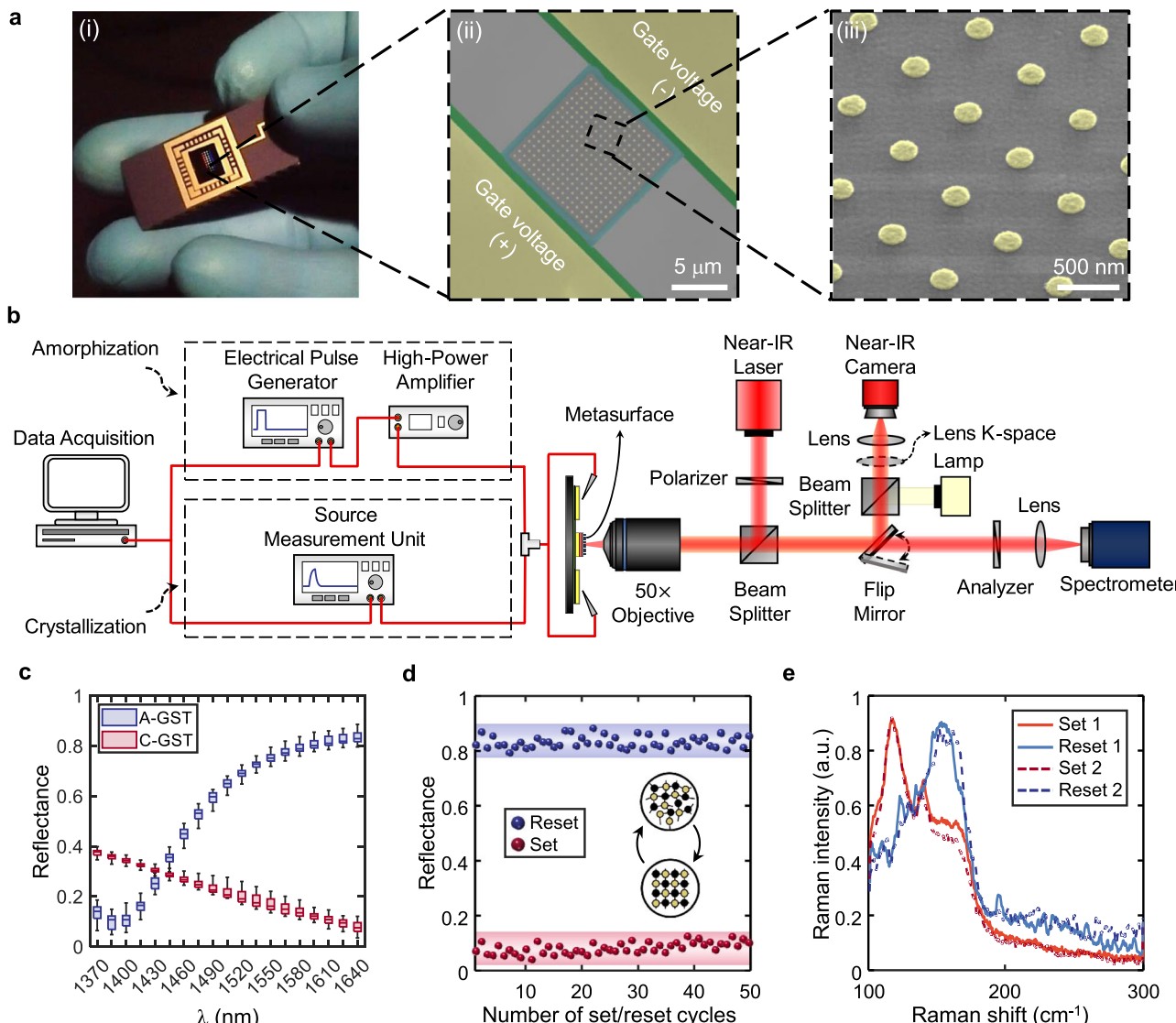

**Fig. 2 Experimental characterization of the electrically reprogrammable metasurface. a** (i) Image of the fabricated sample mounted on a ceramic chip carrier, (ii) tilted false-colored SEM image of the meta-switch comprising the microheater and the phase-change metasurface, and (iii) the magnified bird's eye view of the meta-atom array. **b** Schematic of the experimental near-IR reflectometry and the back-focal-plane imaging setup coupled to signal generators for electrical programming and optical characterization of fabricated devices (see "Methods" for details). **c** Demonstration of the binary operation of the meta-switch: statistical distribution of change in measured reflectance over 50 consecutive cycles of crystallization (red boxes) and amorphization (blue boxes) for 19 equal-distant wavelengths. The lower/upper quartile and the median of the collected data are represented by color-coded lines while the minimum/maximum extent are displayed with black line. The lens K-space is introduced for the beam steering measurements. **d** Cyclability plot of the optical reflectance of the meta-switch during multiple electrical set (red dots leading to C-GST) and reset (blue dots leading to A-GST) pulses. The 95% confidence intervals (shaded areas) of ±1% and ±7.5% are measured for the reflective and absorptive states, respectively. **e** Raman scattering spectra of the GST film after applying two consecutive cycles of set/reset electrical pulses. The A-GST spectra (bluish lines) pose a rather broad peak which are transformed to the dual-band peak ones upon transition to C-GST (reddish lines). The structural parameters of the studied metasurface are $p = 600$ nm, $d_{Au} = 190$ nm, $t_{Au} = 35$ nm, $t_{GST} = 40$ nm, and $t_{Al_2O_3} = 10$ nm.

in the $x$–$z$ cross section of a meta-atom in Fig. 3c show excitation of the SR-SPP mode and LR-SPP mode for $\lambda_1 = 1407$ nm and $\lambda_2 = 1600$ nm, respectively. The two SPP modes do not exhibit the same degree of localization and enhancement. For the former case, the magnetic field is distributed along the interface of the Au backreflector and the bottom $Al_2O_3$. For the latter, the magnetic field is strongly enhanced underneath the nanodisk due to the anti-symmetric current distribution in the two Au parts. The electric field magnitude profile in Fig. 3c (and flowlines of the Poynting vector in Supplementary Fig. S8a) implies that a good portion of the incident energy is dissipated after funneling of the incident wave into the lossy GST film at $\lambda_1$. In contrast, the

coupling of accumulated charges at both lateral end-faces of the nanodisk can form a pronounced electric dipole resonance at $\lambda_2$. Such a strong resonance fairly traps the major energy of the incident light near the nanodisk that is dissipated due to the lossy nature of Au (also see flowlines of the Poynting vector in Supplementary Fig. S8b).

**Phase-change gradient metasurface for dynamic beam steering.** In addition to the active control over the amplitude and resonance features of the reflection spectrum, the studied hybrid platform offers tuning over the phase response of the scattered field enabling dynamic wavefront engineering in the near-IR

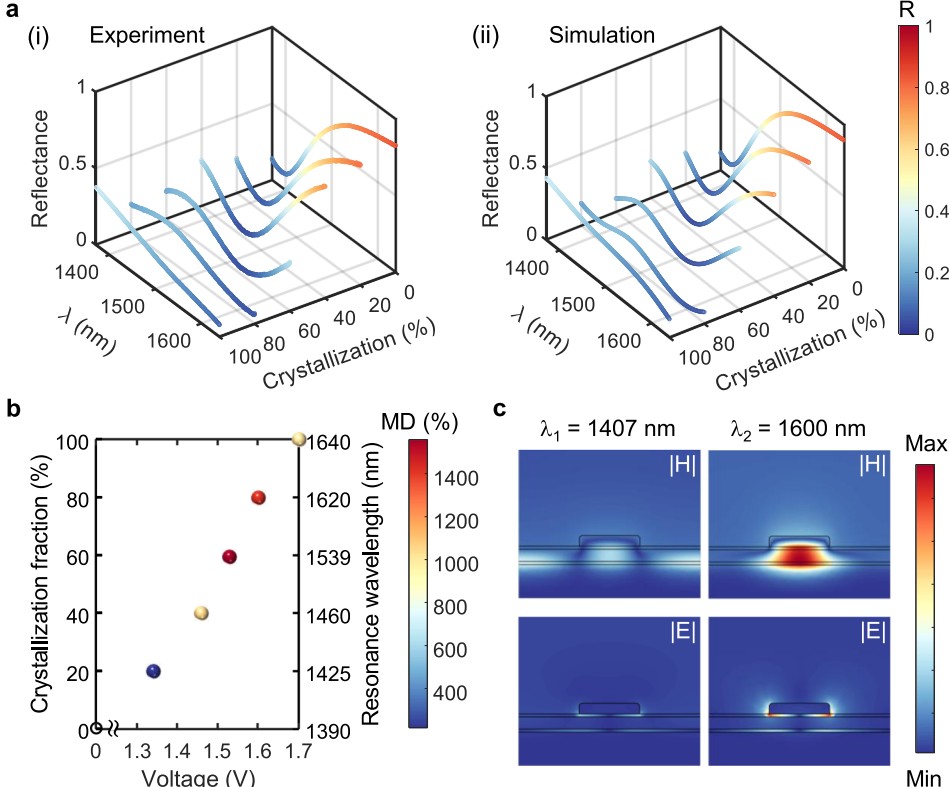

**Fig. 3 Multi-state operation of the active phase-change metasurface. a** (i) Measured and (ii) simulated color-coded reflectance spectra of the programmed meta-switch with A-GST (i.e., 0% crystallization fraction), C-GST (i.e., 100% crystallization fraction), and 4 accessed intermediate states (with 20% crystallization steps). **b** Correlation of the modulation depth (MD), undergoing approximate crystallization fraction, and fundamental resonance wavelength shift with the applied pulse voltage. **c** Inspection of the normalized magnetic field magnitude and electric field magnitude at the resonance wavelengths of the meta-switch with 80%-crystalline GST (in panel (**a**)(ii)) in the x–z plane of a meta-atom.

spectral range. Here, we demonstrate a reconfigurable phase-change gradient metasurface, called meta-deflector, to selectively steer the incident light to the +1st/0th diffracted order when GST is in its amorphous/crystalline state.

The meta-deflector is formed by a 2D array of supercells each consisting of a linear arrangement of 7 meta-atoms (as shown in Fig. 4a) following the working principle of phase-gradient metasurfaces[39]. By monolithically increasing the diameter of Au nanodisks, the SR-SPP mode supported by the metasurface with A-GST undergoes a gradual spectral shift, and thereby ~45° phase shift is progressively added per meta-atom to the reflected wavefront (see Fig. 4a(i)). The more than 315° optical phase coverage achieved upon this evolution is enough for several phase-based optical functionalities, though this comes at the cost of variation in the reflectance on account of the on-resonance operation of the meta-atoms at $\lambda = 1495$ nm (see Fig. 4a(ii)). As a consequence of full crystallization, all designed meta-atoms serve as similar phase retarders, with ~0° phase shift between neighboring meta-atoms, providing uniform scattering efficiency alongside the supercell (see Fig. 4b(i)). The rather flat nature of the reflection amplitude and phase curves in Fig. 4b(ii) stems from the off-resonance characteristics of the LR-SPP mode at the operational wavelength.

Following the generalized Snell's law, we design a meta-deflector such that it enables the steering of light beam to the angles of 20° and 0° in the case of A-GST and C-GST, respectively. The simulated deflection intensities as a function of the steering angle, displayed in 2D maps in Fig. 4c(i), c(ii), are in good agreement with the calculated angles from the theory. The switching contrast ratio, defined as $I_{A-GST}^{20°} I_{C-GST}^{0°} / I_{A-GST}^{0°} I_{C-GST}^{20°}$,

where $I$ represent the reflection intensity, is calculated to be 10.8 dB, which is comparable to state-of-the-art reconfigurable beam-steering metasurfaces[4,5]. A rather non-dispersive behavior is also observed in the angular reflection responses for the spectral range of 30 nm around the designed wavelength (i.e., $\lambda = 1495$ nm) that facilitates the device operation in the S band. The steering capability of the meta-deflector is experimentally validated by fabrication of a 100 μm × 100 μm phase-change metasurface (see Fig. 4d) electrically tunable using the Joule heating process (see Supplementary Information Note 1 and Fig. S9 for details). The far-field radiation patterns for 3 different wavelengths presented in Fig. 4e justify anomalous to specular reflection operation upon switching the state of GST from amorphous to crystalline. It should be mentioned that in the amorphous case, although under normal illumination the deflected beam is mainly coupled to the +1st order, still the 0th order carries a noticeable portion of the reflected energy. We anticipate that the adoption of an enlarged-aperture meta-deflector comprising optimized meta-atoms with free-form geometries coupled to different states of GST can further improve the optical performance of these phase-change gradient-metasurfaces for on-demand beam forming applications such as varifocal lensing.

**Performance analysis using machine learning.** Beyond formal modeling of any physical phenomenon, exploratory analysis with diagrammatic representations is a powerful tool that helps inferring by visualization of the data. The key concept is to form an easy-to-interpret low-dimensional representation of the structured data with the end goal of unveiling data points with unusual attributes, demystifying the underlying connections, and revealing the governing patterns. In this regard, to gain an

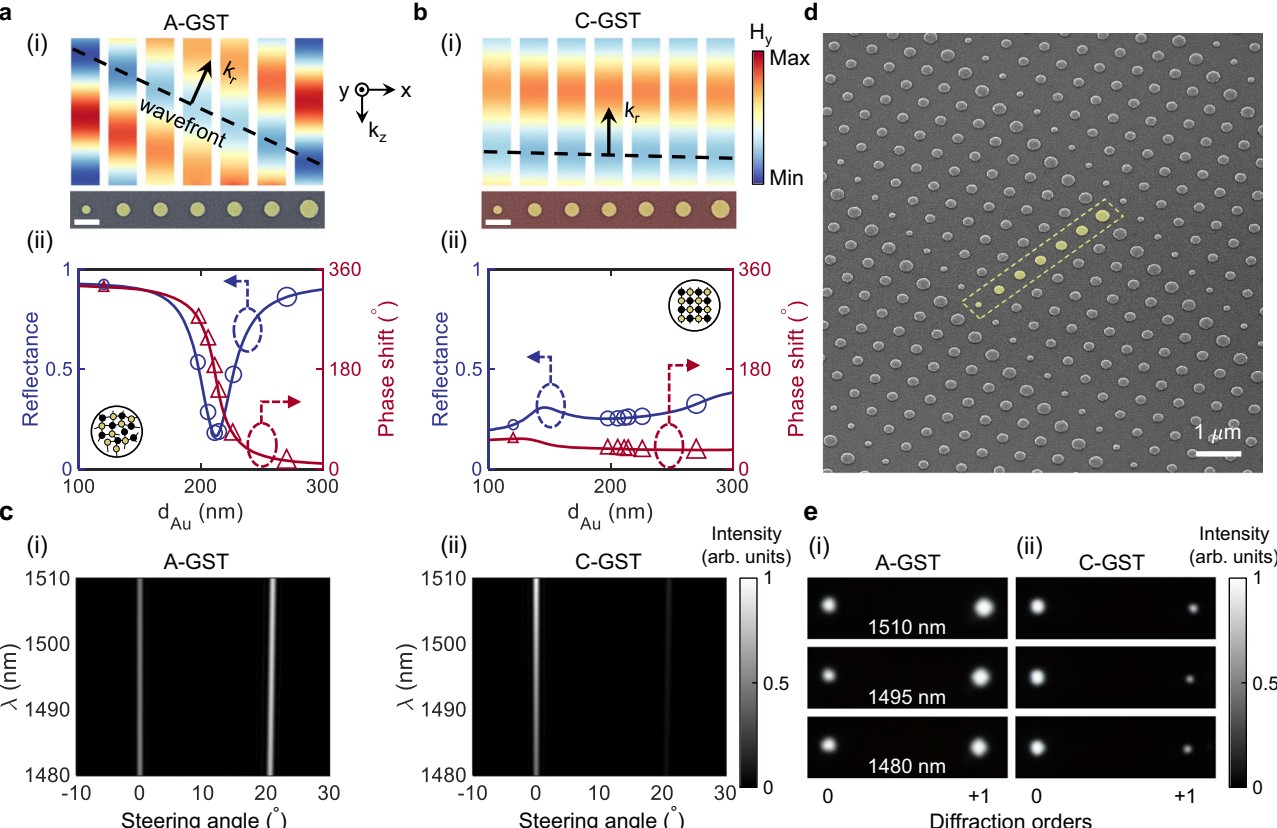

**Fig. 4 Demonstration of an electrically driven dynamic phase-change meta-deflector.** Wavefront evolution from a supercell of the meta-deflector with **a** A-GST and **b** C-GST. (i) Simulated scattered magnetic fields from 7 constitutive meta-atoms (with diameters $d_{Au}$ of 120, 196, 206, 212, 216, 224 and 270 nm and $p = 620$ nm) of the supercell at the same time instant. The phase shift increment induced by neighboring meta-atoms can shift the position of the peak of the field up to a wavelength (as shown by the black dashed line) in the amorphous state. Each strip is calculated by an independent full-wave numerical simulation of a meta-atom illuminated by a x-polarized plane-wave with free-space wavelength $\lambda = 1495$ nm. The scale bar in the inset is 400 nm. (ii) Reflectance (left axis) and phase shift (right axis) of the scattered field from the meta-atom as a function of the Au nanodisk diameter. Discrete amplitude and phase values associated with each meta-atom in (i) are defined on the curves. **c** Simulated normalized deflection intensities from the dynamic meta-deflector as a function of the steering angle and wavelength for the cases of (i) A-GST and (ii) C-GST. **d** Angled SEM image of the fabricated sample with false-colored Au nanodisks in a supercell defined by the dashed box. **e** Measured normalized far-field radiation intensity of the active meta-deflector upon conversion of the structural phase of GST from (i) A-GST to (ii) C-GST using the Joule heating process. Images are obtained from the CCD camera in Fig. 2b for three incident wavelengths.

intuitive understanding of the overall performance of the meta-switch, without relying on the apriori knowledge, we leverage data visualization using an unsupervised machine learning approach. This method transforms a set of high-dimensional data sets (e.g., high-resolution reflectance spectra from the meta-switch with A-GST and C-GST) into low-dimensional maps while preserving necessary information (e.g., nature of the governing mode)[40]. However, this should not be translated as the ability of data visualization techniques to reveal underlying physics of light-matter interaction or extract physical information. Among several options[41], we leverage a nonlinear dimensionality reduction technique called t-distributed stochastic neighbor embedding (t-SNE) extensively used for data exploration and visualization of high-dimensional data in image processing[42]. The t-SNE algorithm aims to match neighbors in a higher-dimensional space to a lower-dimensional one by measuring the similarity between pairs of variables. It then optimizes these two similarity measures based on a predefined cost function. Upon applying to a high-dimensional but well-clustered data set, t-SNE tends to generate a visual embedding with distinctly isolated clusters.

Figure 5a represents three-dimensional (3D) embeddings of reflectance responses of the metasurface in Fig. 1a with different structural parameters for both A-GST and C-GST cases.

Numerical simulations are carried out in the operational spectral range, i.e., 1370–1640 nm, for 3600 different metasurfaces with a wide range of randomly selected values for the structural parameters (i.e., $p$, $d_{Au}$, $t_{GST}$, and $t_{Al_2O_3}$). The implication of the formation of the two distinguishable unfolded clusters corresponding to A-GST (blue points) and C-GST (red points) in the 3D latent space is twofold. Incorporation of GST in the meta-atom grants a different class of responses not easily accessible through just variation of structural parameters with one GST state. Accordingly, the GST crystallization state can be considered as an effective tuning knob to modify the metasurface performance. Furthermore, the metasurfaces with A-GST and C-GST are likely governed by modes with distinct natures (see Supplementary Note 6 for more discussions). To elucidate the results, the reflectance spectra for metasurfaces with 2 specific sets of structural designs, (i) $d_{Au} = 240$ nm with varying $p$, and (ii) $p = 500$ nm with varying $d_{Au}$ (see corresponding SEM images in Fig. 5b, d) are shown in Fig. 5c, e, respectively. The responses are also included in the 3D dimensionality-reduced space in Fig. 5a using color-coded shapes. It is seen that for case (i), the corresponding color-coded circles are fairly centered in the 3D embeddings while for case (ii), the magenta shapes are distinguishably expanded over the 3D latent space. Looking into

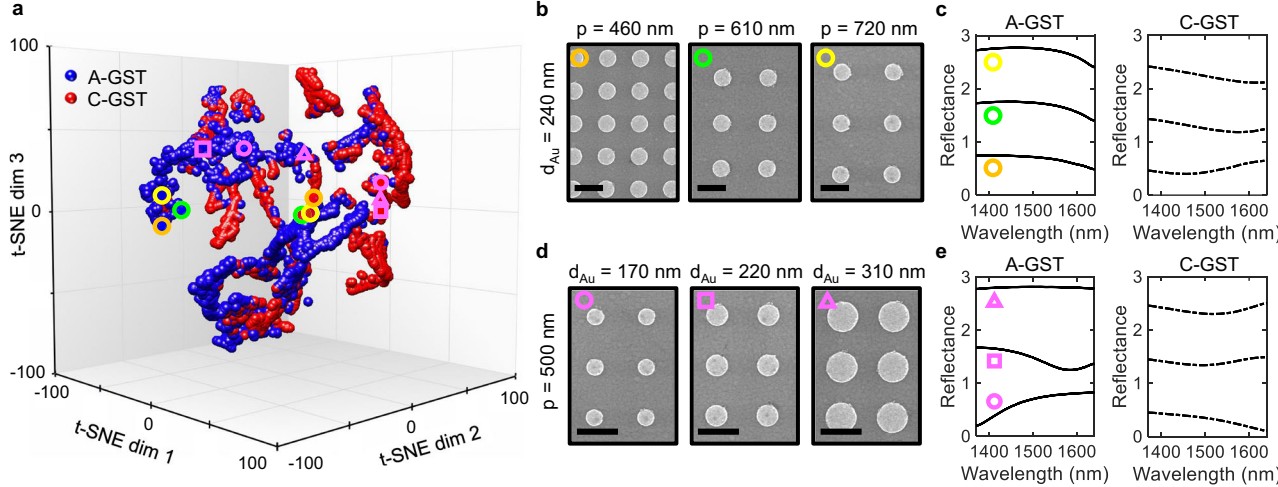

**Fig. 5 Performance analysis of the reconfigurable metasurface using machine learning. a** Unsupervised 3D nonlinear embedding of the simulated reflectance spectra of the phase-change metasurface in Fig. 1a with different structural parameters (i.e., $p$, $d_{Au}$, $t_{GST}$, and $t_{Al_2O_3}$). The embeddings corresponding to A-GST and C-GST are depicted as blue and red points, respectively. **b, d** Top-view SEM images of the fabricated meta-switches with the experimental data marked in (**a**) and their corresponding reflectance responses in (**c, e**), respectively. The scale bars in all SEM images indicate 400 nm. The reflectance curves in (**c, e**) are shifted along the vertical axis for the sake of clarity.

Fig. 5c, e, the reflectance spectra for the latter case, specifically for A-GST, evolve more rapidly than that for the former case.

While the t-SNE algorithm provides a helpful visualization of the cluster of responses granted by each state of GST, it is not straightforward to use it to compare the importance of the different design features in varying the optical response. Such information is crucial in various aspects: (i) it provides valuable insight about the robustness of switching operation against variation of each design parameter, (ii) it can be used to devise non-uniform sampling of the overall range of different design parameters to form the optimum library of dataset considerably reducing the computation requirements, and (iii) it identifies the most vulnerable parameters to the fabrication errors to help in customizing the optimal fabrication process. To enable the benefits of ranking the importance of design parameters in achieving the maximum modulation depth at 1550 nm (with 10% bandwidth) upon switching from A-GST to C-GST, we employ a feature-selection algorithm, namely the wrapper method[43]. Starting from an empty feature subset, the algorithm sequentially adds each of the structural parameters as a candidate to the subset and performs cross-validation by repeatedly calculating the evaluation criterion until the stopping condition is reached. By implementing the wrapper algorithm on the training dataset of our metasurface structures, we find that the most influential design parameters (in the high-to-low order) are $t_{GST}$, $d_{Au}$, $p$, and $t_{Al_2O_3}$ (see Supplementary Note 8 for more details). Note that this conclusion is made through the learning process without adding any physical apriori knowledge. We also support these inferences through studying the evolution of underlying modes of the metasurface in the near-IR regime as discussed in Supplementary Note 9. From the experimental point of view, the influential role of GST thickness on the modulation depth demands employing of reliable PCM growth techniques such as atomic layer deposition with high precision and uniformity.

In summary, we demonstrated chip-scale electrically driven phase-change metadevices through the incorporation of a heterostructure microheater integrated with a hybrid plasmonic-PCM metasurface. A systematic design approach by leveraging a multiphysics electrothermal study and an electromagnetic analysis of the metadevice was presented. Without comprising the optical performance, our introduced platform is capable of quasi-continuous, reversible, and non-volatile tuning of the fundamental modes of the meta-atom, i.e., LR-SPP and SR-SPP. An active meta-switch with an unprecedented optical efficiency reaching 80%, unparalleled electrical modulation of the reflectance by more than eleven-fold, unmatched spectral tuning over 250 nm, and potential operation speed of a few kHz was experimentally demonstrated. In addition, we experimentally demonstrated the active beam steering functionality in the near-IR spectral range by leveraging an electrically actuated phase-change gradient metasurface. By switching the state of GST from amorphous to crystalline, we could selectively control the amount of the optical power concentrated into the +1st and the 0th order of diffraction. Furthermore, we leveraged data-driven machine learning approaches to study the effect of GST on expanding the attainable response space of a reconfigurable metasurface and rank strategical structural parameters influencing the overall optical response of the meta-switch. Our findings can open new directions for reconfigurable flat optical devices manipulating the dominant properties of light for various applications including imaging, computing, and ranging.

## Methods

**Electrothermal simulations.** The COMSOL Multiphysics software package is used for Joule heating simulations and consequently transient thermal behavior study of the electrically driven metadevice. The configuration of the microheater is carefully chosen to meet the design specifications within the limitation of testing equipment. A coupled multiphysics model including the Electric Currents module, for simulating the electrical current profile, and Heat Transfer in the Solid module, for calculating the heating exchange and temperature distribution, is employed. In the Heat Transfer module, infinite element domains are considered for the side boundaries of the constructed model. In addition, the top and bottom surfaces are given a convective heat flux boundary condition with ambient temperature of $T = 20\,°C$. The effect of large Au pads, facilitating the engagement with high-frequency probes or wire bonding to an external board is considered in our 3D model (see Supplementary Fig. S2). The parameters of contributed materials are taken from four-point probe measurements and existing experimental data in the literature (see Supplementary Note 1 for details).

**Numerical simulations.** Full-wave simulations are carried out using the commercial finite element method (COMSOL Multiphysics) and verified by the finite integral technique software CST Microwave Studio. We model the meta-atom of the metasurface by considering periodic boundary conditions on the vertical sidewalls along the $x$ and $y$ directions. We apply perfectly matched layers at the truncated air boundary along the $z$-direction to avoid spurious back reflections. In our simulations, a broadband propagating plane wave perpendicularly (unless

otherwise stated) is launched toward the metasurface from free space. A 2D monitor is used in the free space above the metasurface to record the reflected light amplitude and phase. The refractive index of $Al_2O_3$ and GST used in the simulations are obtained through spectroscopic ellipsometry measurements (see Supplementary Note 2 and Fig. S5). The optical constants of Au is obtained from experimental values reported in ref. [44].

**Theoretical modeling**. Upon scattering of the incident light by the nanodisks array, the in-plane component of the wavevector, i.e., $k_{||}(\lambda)$, matches that of the LR-SPP mode whose dispersion can be described by Bragg's equation[45]

$$\mathbf{k}_{||}(\lambda) \pm i\mathbf{G}_x \pm j\mathbf{G}_y = \mathbf{k}_{LR-SPP}(\lambda),\qquad(1)$$

where $k_{||}(\lambda) = k \sin(\theta)$, in which $\theta$ is the angle of incidence with respect to the normal direction (z in Fig. 1a(i)), $|\mathbf{G}_x|$ and $|\mathbf{G}_y|$ ($|\mathbf{G}_x| = |\mathbf{G}_y| = 2\pi/p$) are the Bragg vectors associated with the two orthogonal lattices of the metasurface, i and j are the integers accounting for the orders of the scattering event, and $\mathbf{k}_{LR-SPP}(\lambda)$ is the wavevector of the LR-SPP mode. White dashed lines in Supplementary Fig. S6a represent the normally incident light coupled to $(i, j) = (1, 0)$ LR-SPP mode of the phase-change metasurface. Indisputably, the simulated reflectance dip in Supplementary Fig. S6a fairly follows the trend of dashed lines as p increases, which corroborates the LR-SPP mode of the lower wavelength mode. The slight discrepancy can be ascribed to the high refractive index of GST that fundamentally limits the excitation of higher diffraction orders, as a source of adding extra momentum to the incident light to excite the LR-SPP mode, in the intermediate layers. The reflectance maps in Supplementary Fig. S6b display that, for all three phases of GST, variation of $d_{Au}$ negligibly affects the location of the resonance dip, as predicted by Eq. 1, which ascertains the existence of the LR-SPP mode in the lower wavelengths.

Along with the LR-SPP mode, when the phase-change metasurface is illuminated by an x-polarized beam, the nanodisk, and subjacent layers can be modeled as a Fabry–Pérot resonator supporting the SR-SPP mode with a resonantly enhanced field at the interface of the Au nanodisk and the top $Al_2O_3$ layer. Such a highly confined mode is excited in virtue of constructive interference of propagating waves between the two lateral end-faces of the nanodisk. By encountering the two truncations of the resonator, the SR-SPP mode is partially scattered into free-space modes and partially reflected back at each interface. To satisfy the resonance condition, the round trip accumulated phase must be equal to an integer multiple of $2\pi$ yielding[46]

$$d_{Au,m} = \frac{m\pi - \phi(n_{eff})}{2\pi}\lambda_{SR-SPP},\qquad(2)$$

where $d_{Au}$ is the diameter of the nanodisk, m is a positive integer indexing the resonance order, $\lambda_{SR-SPP}$ is the wavelength of the SR-SPP mode, and $\phi(n_{eff})$ (in which $n_{eff}$ is the effective refractive index of the ambient) is the phase induced upon reflection of the SR-SPP mode at the two end-faces of the nanodisk. The dependence of $\lambda_{SR-SPP}$ on $d_{Au}$ is represented with the black dashed lines in Supplementary Fig. S6b for A-GST, P-GST, and C-GST. We attribute the longer wavelength mode to the SR-SPP as evidently the location of the resonance dip is in accordance with the predicted ones from theoretical calculations. Additionally, negligible dependence of the reflectance dip to the variation of p, as illustrated in reflectance maps in Supplementary Fig. S6a, renders the localized nature of this mode.

Evident from Supplementary Fig. S6a, in the case of A-GST with low intrinsic loss, LR-SPP and SR-SPP are spectrally distant. Partial crystallization of GST draws the slightly broadened modes to the center of the telecom spectral window, where they fairly overlap. By fully converting the state of GST using electrical Joule heating, a significant index contrast can be observed (see Supplementary Fig. S5), which further broadens and dampens the existing resonance modes. In this regard, dynamically tunable hybrid plasmonic-PCM metasurfaces offer high potentials for engineering both the amplitude and phase properties of incident light waves enabling switching and beam steering applications.

**Sample preparation**. The electrically driven reprogrammable metasurfaces are implemented through a series of standard and customized fabrication processes (see the flow diagram in Supplementary Fig. S1). We start with the atomic layer deposition (ALD) of a 100 nm-thick $HfO_2$ layer on a 500 µm-thick Si substrate to prevent direct contact between the microheater and probing pads and the base substrate. Next separate steps involve electron beam (e-beam) lithography to define the patterns of the microheater/probing pads followed by RF sputtering of a 50 nm-thick W layer/e-beam evaporation of a 100 nm-thick Au layer and ultimately a lift-off process. Then, e-beam lithography is performed to define the aperture on the microheater where the metasurface is finally located. Sequential depositions of a 20 nm-thick $Al_2O_3$ layer by ALD, an 80 nm-thick Au layer by e-beam evaporation, and a 10 nm-thick $Al_2O_3$ layer by ALD are performed to fill the opening. This follows by the deposition of a 40 nm-thick GST layer from a stochiometric target in an RF sputtering system and subsequent deposition of a 10 nm-thick $Al_2O_3$ as a capping layer in the ALD chamber. After the lift-off process, spin coating of a thin layer of polymethyl methacrylate (PMMA) is performed and Au nanodisk arrays

are lithographically defined and formed by developing in a room-temperature methyl isobutyl ketone/isopropyl alcohol (MIBK/IPA) mixture. As the last step, e-beam evaporation of a 35 nm-thick Au layer is carried out followed by an overnight lift-off process. An ultrathin layer of Titanium (Ti) is used as the adhesion for Au.

**Material characterization**. The complex refractive indices of A-GST and C-GST with different thicknesses are calculated using spectroscopic ellipsometry measurements with three different angles of incidence (50°, 60°, 70°) over 1000–2000 nm spectral range (see Supplementary Fig. S5 for the thickness used in this work). Tauc-Lorentz and Cody-Lorentz[47] are chosen as fitting models with optical bandgap, oscillator width, Lorentz oscillator amplitude, resonance energy, and Urbach energy as fitting parameters. We consider the uniform thickness shrinkage of GST (~5%) upon full crystallization (see Supplementary Fig. S15) in our calculations. The surface roughness measured by the atomic force microscopy measurement (see Supplementary Fig. S16) is also used in our calculations. To verify the material state of GST, we benefit from confocal Raman microscopy to study the Raman scattering of the A-GST/C-GST film after applying set/reset pulses. The power of the primary laser is set at low values to prevent crystallization during measurements. As shown in Fig. 2e, the normalized Raman spectra for the two randomly chosen conversion cycles exhibit a similar trend; possessing a rather broad peak in the amorphous state and a dual-band peak upon transition to the crystalline state. X-ray photoelectron spectroscopy is performed to study the existing elements and determine the binding energies of the core electrons. Core-level spectra of elements are plotted in Supplementary Fig. S17 obtained through a survey scan within the binding energy range of 0–600 eV. We also leverage X-ray crystallography to determine the atomic structure of GST in its extreme phases. The corresponding X-ray diffraction patterns in Supplementary Fig. S18 show Bragg peaks verifying the face-centered cubic configuration of C-GST.

**In situ electrical characterization**. In our experiments, full crystallization and amorphization processes for the meta-switch are performed by applying a 1.7 V set pulse with 200 µs-long double exponential waveform and a 3.8 V reset pulse with 200 ns-long rectangular shape, respectively, to the meta-switch. For the former, the voltage pulses features zero width and leading/trailing edge of 100 µs resolution imposed by the limitations of the source measurement unit (Keithley 2614B). Electrical pulses with lower peak voltages than that of the set pulse are also used to transform the state of GST between its extreme phases in multiple states. The reset pulse has a leading/trailing edge of ~10 ns that is generated by Tektronix AFG3252C function generator and delivered to ENI 510L RF power amplifier before applying to the device. The short pulse used in the latter biasing scheme avoids unwanted material flow during amorphization. Small differences with simulated results are mainly attributed to the discrepancy between the thermal properties of fabricated and simulated materials, the parasitic resistances associated with the probing pads and contacts, random resistance variation of the W patch, and the thermal boundary resistance between contributed materials. For the full crystallization of the GST film in the meta-deflector, we apply a a 10.5 V set pulse with 200 µs-long double exponential waveform to the Au pads connected to the microheater.

**In situ optical measurements**. Experimental optical measurements is performed by directly measuring the intensity of the reflected light from the surface of the fabricated device installed on a xyz-translation stage (see Fig. 2b). A low-power beam (to prevent the conversion of GST during measurements) from a fiber-coupled light source is focused on the device using a ×50 Apochromatic near-IR objective lens with numerical aperture of 0.42. For the case of the meta-switch, the focal spot is ~8 µm in diameter measured using the knife-edge technique. To measure the reflected signal a beam splitter is installed right before the surface of the sample to allow separation of the incident and the reflected signals. Normalization is done by dividing the intensity of a reference beam with the same spot size reflected from a smooth surface of an Au patch fabricated near the metadevice under test. To visualize the device under test, a second beam splitter is used in the optical path to direct the reflected visible light to a near-IR charge-coupled device camera. Co-located in situ optical and electrical measurements are carried out while the metadevice under the microscope is connected to the external signal generators with a high frequency Infinity probe. By inserting and removing a lens K-space near the CCD (see Fig. 2b), we can switch between the real space and the Fourier plane for the back-focal-plane imaging of the objective lens.

## Data availability
Relevant data supporting the key findings of this study are available within the article and the Supplementary Information file. All raw data generated during the current study are available from the corresponding author upon reasonable request.

## Code availability
The codes that support the plots within this paper are available from the corresponding author upon reasonable request.

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

## Acknowledgements

The work was primarily funded by Office of Naval Research (ONR) (N00014-18-1-2055, Dr. B. Bennett) and by Defense Advanced Research Projects Agency (D19AC00001, Dr. R. Chandrasekar). W.C. acknowledges support from ONR (N00014-17-1-2555) and National Science Foundation (NSF) (DMR-2004749). A. Alù acknowledges support from Air Force Office of Scientific Research and the Simons Foundation. M.W. acknowledges support by the Deutsche Forschungsgemeinschaft (SFB 917). M.E.S. acknowledges financial support of NSF-CHE (1608801). This work was performed in part at the Georgia Tech Institute for Electronics and Nanotechnology (IEN), a member of the National Nanotechnology Coordinated Infrastructure (NNCI), which is supported by NSF (ECCS1542174).

## Author contributions

S.A. conceived and designed the study, carried out the modeling of metadevices, conducted the device fabrication, and performed material, optical, and electrical characterizations. O.H. and H.T. contributed to the device fabrication, material characterization, and multi-state metasurface measurements. M.T. and W.C. assisted with the development of the optical setup and measurements. A.K., A.A.E, and A. Alù helped with the optical and electrothermal analysis. C.T., S.D., E.P., and M.W. optimized and deposited the phase-change material for the metadevices. M.E.S. helped with the material characterization. S.A. wrote the manuscript with input from all authors. All authors discussed the results and commented on the manuscript. A. Adibi supervised the whole project and revised the last version of the manuscript.

## Competing interests

The authors declare no competing interests.
