## [Peer Review File · Nature Communications]

REVIEWER COMMENTS

Reviewer #1 (Remarks to the Author):

This work is interesting research and an important step toward electrically switchable metasurfaces. Results are well presented and experimental and theoretical data agree well. I support its publication and I have several comments summarized below:

1. What is the beam diameter used for optical illumination of the metasurface?
2. What is the spatial distribution of the reflected beam for two different phases of GST? I expect due to the temperature gradient there should be some nonuniformity in reflection across the metasurface area. The authors should have this data since the measurement setup includes an IR Camera.
3. From a theoretical point of view (electrothermal analysis at $\sim 1\text{kHz}$) how big the area of metasurface can be before there is a large spatial nonuniformity in reflection value (e.g., 20% modulation, i.e. across different locations of the metasurface for example from the center region to the edge of metasurface)?
4. Were all reflection measurements performed under electrical pulse (as described in Method) or sometimes DC-voltage/long-pulses used to switch between the different phases of GST?
5. The efficiency of presented metasurfaces is $\sim 80\%$ for amplitude modulation and drops when the metasurface is used for phase modulation. Can authors comment on what can be done to improve the efficiency in the phase modulation scheme which represents more applications for metasurfaces?

Reviewer #2 (Remarks to the Author):

Abdollahramezani and his co-workers present a phase-change material (PCM) based meta-reflector, that is electrically tunable, allowing around 10-fold reflectance change in the infrared spectral range around 1.5microns. The device is based on a micro-heating platform to reproducibly control the phase transition of a GST layer, on top of which plasmonic nanostructures are placed. The authors claim that they reach record high reflectance of $\sim 80\%$, tunable by a factor of ~ 10 with very high switching speed and the capability to controllably address intermediate switching states. The paper is well illustrated and the results are mostly convincing (with the exception of the last figure, see below). However, I find the structure of the manuscript not too clear, interesting and important analysis of the physics is out-sourced to the supplementary pdf and the discussion remain often quite superficial. My main concern is however the novelty over several recently published, similar results, which should be presented more clearly. These and other points that should be addressed are detailed below. I recommend to either perform a major revision to render the paper impactful enough for a broad-audience high impact journal like Nat. Comms. Alternatively the authors could submit their work to a less ambitious journal upon minor revisions of the unclear parts.

- My main concern is the novelty, mostly with respect to references [31,32], since the platform seems quite similar. The authors mainly seem to state that their reflectance is higher, due to less mechanical deformation and a somehow varied material stack. I feel that this is rather little novelty for a high impact journal such as Nat. Comms. I feel that for example an experimental demo of the deflection switching device, simulated in the SI Fig S6 would represent an appealing novelty.

- how does the presented platform compare to more traditional tunable reflectance devices such as liquid crystal based approaches?

- It seems that the platform does not allow control over a full 2π phase cycle, in the SI 180 degrees are demonstrated, but maybe this can be combined by hybridization with some localized surface plasmon or so to go beyond 1π . Can the authors comment on that?

- In the introduction, the authors state that their platform is the first fully reversible reconfigurable GST-based metasurface. To my knowledge other works claim to achieve also fully reversible GST metasurfaces [e.g. Julian et al, *Optica* 7(7) 746-754, 2020], but also [31] and [32]. the authors should lower their claims accordingly.

- What are the differences of the here presented micro-heaters in comparison to what is commercially developed for phase-change RAM applications?

- Would the approach work also in a pure layer-stack geometry (thus without the plasmonic nano-discs on top), similar to recently reported radiative cooling applications [e.g. Raman et al. *Nature* 515, 540-544 (2014)]?

- The usefulness of the section "Performance analysis using machine learning" is imo quite doubtful. In this context the authors state that the dimensionality reduction is "preserving necessary information (e.g., nature of the governing mode)", which falsely might sound as if the unsupervised learning would automatically extract physical information. This is however not the case, it just clusters the data into some kind of principal components, regardless the underlying mechanism. Linking the latent space to physics can be actually highly non-trivial, the discussion here can be misleading in this regard. It can also mislead the reader to state that the fact that A- and C-GST points span the full 3D t-SNE latent space has some physical meaning about a broad coverage of different possible reflection responses through the tested designs. t-SNE is actually built such that it tries to distribute clusters over the full target latent space. Using for instance only c-GST would probably also lead to a more or less fully covered latent-space. The authors might look into [Kiarashinejad et al. *Advanced Intelligent Systems* 2, 1900132, 2020] for an example how to use dimensionality reduction to quantitatively assess the feasible physical responses of an optical system.

- In the very last paragraph the authors mention a "wrapped algorithm". It is not clear at all to me how this algorithm works and how the authors come to the mentioned ranking of highest variability design parameters. Also, after presenting the results, the authors don't discuss the meaning of this ranking like the consequences for design optimizations or implications concerning fabrication robustness.

- In general I find the recent works on PCM based switchable meta-reflectors ([31, 32] and the present one) really interesting and this seems to open exciting routes for further concepts. As a suggestion: I would be very interested in a discussion about the possibility to micro- or even nano-structure heating elements in order to induce a spatially heterogeneous crystallization / amorphization pattern. The first question is probably, whether this is feasible from a fabrication point of view as well as regarding the electronic engineering aspect (micro-electrode array with very dense connections to the metasurface area). The second point would be the local heating: Could it be possible to induce the phase change in a controlled way on a (sub)-micron length-scale, without affecting neighboring "phase-change cells"?

- Personally I find the way figure 3a presents the reflectance spectra not very helpful. It would be far easier to compare the spectra quantitatively if they were plotted in a 2D plot, each spectra offset by a constant increment (similar to for example fig 4c and e).

- in the supporting information figure S7, the reproducibility is indeed clearly demonstrated, it would just be nice to plot the initial or average reflectance in every figure as a baseline for better comparison.

Reviewer #3 (Remarks to the Author):

The authors demonstrate a metasurface that is dynamically and reversibly switchable between reflecting and absorbing functionalities in the NIR. They demonstrate significant performance improvements in terms of switching speed and efficiency over similar designs that have been previously reported. The results are highly significant to the field of reconfigurable optics and were obtained via scientifically sound methodology. I believe the manuscript could be improved if the following are addressed:

1. It is suggested that a PCM device can maintain performance for $\sim 10^{12}$ cycles, but it is not stated what is achieved for this device (only data up to 50 is presented). Even after only 50 cycles there is a noticeable upward drift in the reflectance in the C-GST state. How many cycles are achievable and what causes the limitation on this number?
2. Tuning of the P-GST states is characterized as quasi-continuous. What is the limit on the resolution of which states are achievable and how many states, in principal, could be accessed?
3. Is there any indication as to the nature of the t-SNE dimensions of the latent space? It is evident that A-GST and C-GST structures access different regions within this space, but it is not apparent how to interpret a given point and relate it to the physical response of the metasurface. Particularly in regards to the statement: "...incorporation of GST in the meta-atom considerably spans the attainable responses not accessible through just variation of structural parameters with one GST state.", how are the responses attainable only with C-GST meaningfully different from the responses only achievable with A-GST? It seems premature to conclude that the crystallization state is an ideal tuning knob, if it's not understood what exactly is being tuned (i.e., what it means to travel from one point to another within the latent space).

Reviewer #1 (Remarks to the Author):

This work is interesting research and an important step toward electrically switchable metasurfaces. Results are well presented and experimental and theoretical data agree well. I support its publication and I have several comments summarized below:

Our response: We appreciate the Reviewer’s interest in our work and the careful review with a positive evaluation. We found the questions and suggestions raised by the Reviewer extremely helpful for improving the quality of our work. A detailed response to each comment is given below.

- What is the beam diameter used for optical illumination of the metasurface?

Our response: The focal spot is $\sim 8 \mu\text{m}$ in diameter measured using the knife-edge technique. **We have added this information to “In situ optical measurements” Section in Methods.**

- What is the spatial distribution of the reflected beam for two different phases of GST? I expect due to the temperature gradient there should be some nonuniformity in reflection across the metasurface area. The authors should have this data since the measurement setup includes an IR Camera.

Our response: The magnified spatial distribution of the reflected beam from the metasurface in two different states of GST, i.e., C-GST and A-GST, at $\lambda = 1450 \text{ nm}$ have been added as **Supplementary Fig. S2c** (also shown below). These rather uniform profiles across the device are expected since a nonpatterned GST patch with even topography is used, which significantly reduces the surface-induced scattering loss. This evenness is also suggested by the electro-thermal simulations presented in Fig. 2b showing a fairly uniform heat profile across the microheater at the end of the set/reset pulse. In contrast to the A-GST case, where a trace of jags appears at the circumference after amorphization, the C-GST case has a more uniform boundary. This follows the rationale that long nature of the set pulse allows uniform formation and growth of crystalline zones in the background of A-GST; however, the fast reset pulse randomly leaves some partially amorphized zones at the edge of the metasurface far from the center of the heater.

Figure R1: Magnified spatial distribution of the reflected beam from the metasurface shown in Fig. 2a with (i) C-GST and (ii) A-GST at wavelength of 1440 nm. Thanks to the fairly even heat profile generated across the microheater at the end of the set/reset pulses, a uniform spatial distribution is observable for both cases.

- From a theoretical point of view (electrothermal analysis at $\sim 1\text{kHz}$) how big the area of metasurface can be before there is a large spatial nonuniformity in reflection value (e.g., 20% modulation, i.e. across different locations of the metasurface for example from the center region to the edge of metasurface)?

Our response: Spatially uniform heat generation over the whole volume of the GST layer is indeed an important criterion that should be carefully addressed to ensure access to a reliable and repeatable optical performance. The heater size under the metasurface area plays a prominent role in temperature uniformity across the device. We discussed this case for two different sizes of the microheater (with similar size metasurfaces) in Supplementary Note 1 and Figs. S2d-S2g. We have now extended the discussion by exploring a metasurface with a larger size that can tolerate 20% reflectance contrast at 1640 nm. According to Fig. 2, this corresponds to 40% crystallization change between the center (with the full crystalline state) and edges (with 60% crystallization fraction) of the metasurface. In order to comply with this criterion, the temperature variation should be limited to ~ 50 °C between the center and edges, with ~ 260 °C and ~ 212 °C, respectively, of the metasurface (see Supplementary Fig. S2b). Though microheaters with larger footprints improve the temperature uniformity across the device, for the sake of miniaturization, we have considered the area of the microheater 15% larger than the metasurface area (similar to what we demonstrated experimentally). For a better comparison, the multiphysics simulation is carried out under the application of a 200- μ s-long (or equivalently at the speed of ~ 5 kHz) double exponential waveform that necessitates a peak voltage of 6.7 V. Supplementary Fig. S3 (also shown below) displays the temperature distribution in a cross section in the center of the GST film. The isothermal contours reveal a ~ 50 °C temperature variation across an $80 \mu\text{m} \times 80 \mu\text{m}$ metasurface upon applying an electrical pulse with the predefined properties. **We have reflected this complementary discussion to Supplementary Note 1 and Fig. S3 on the design of the microheater.**

Figure R2: Simulated electrothermal model for a large-scale microheater with a $92 \mu\text{m}$ width. The temperature map in the cross section at the center of the GST film is shown. The isothermal contours reveal a ~ 50 °C temperature variation across an $80 \mu\text{m} \times 80 \mu\text{m}$ metasurface at the end of a 200 μs -long 6.7 V set pulse.

- Were all reflection measurements performed under electrical pulse (as described in Method) or sometimes DC-voltage/long-pulses used to switch between the different phases of GST?

Our response: For the full crystallization process, we use a low-voltage set pulse (with a 200 μs -long double exponential waveform and a peak voltage of 1.7 V) that heats up amorphous GST above the crystallization temperature (~ 160 °C). This pulse is sufficiently long that guarantees full nucleation and formation of monolithic crystalline islands without relying on a stationary DC voltage [*Nature materials* 6.11 (2007): 824-832, *MRS bulletin* 39.8 (2014): 703-710]. By

decreasing the voltage amplitude while keeping the pulse length unchanged, arbitrary fractions of amorphous and crystalline states are registered in the material that results in partially crystalline GST. Figure 3b shows the correlation between the measured applied voltage pulse and the crystallization fraction approximated based on the effective-index-medium theory. For the amorphization process, a single high-voltage pulse (with 200 ns-long rectangular waveform and a peak voltage of 3.4 V) featuring a very short leading/trailing edge is employed that rapidly increases the temperature of the crystalline GST above the melting temperature (~ 630 °C) followed by quenching such that GST solidifies in the amorphous state. It is notable that the quenching process with a high cooling rate is essential for the amorphization. This necessitates employing a short electrical pulse for reamorphization. We detailed these processes in the second paragraph of “Electrothermal analysis of the integrated heterostructure metadvice” Section in the main text and “In situ electrical characterization” Section in Methods.

- The efficiency of presented metasurfaces is $\sim 80\%$ for amplitude modulation and drops when the metasurface is used for phase modulation. Can authors comment on what can be done to improve the efficiency in the phase modulation scheme which represents more applications for metasurfaces?

Our response: We would like to clarify that we have considered two different strategies for realization of gradient phase modulation: i) local tuning of the GST cell in each individual meta-atom (Supplementary Fig. S10), and ii) global control over the whole GST film embedded in the metasurface (Fig. 4 in the revised manuscript). For the former case, our criterion is to design a meta-atom that can cover 180° phase shift with minimum reflectance variation upon transition of GST from the amorphous state all the way to the fully crystalline state with 20% crystallization steps (see Supplementary Fig. S10). This enables realization of continuously tunable optical functionalities, e.g., beam steering, holographic imaging, and so on. Since there exists a single SPP mode (pure or hybrid) at each crystallization level governing the optical performance of the metasurface, the optical efficiency is fundamentally restricted due to a strong correlation between the scattering amplitude and phase. On the latter case, **we have dedicated extra efforts to experimentally demonstrate, for the first time to our knowledge, an electrically tunable GST-based gradient metasurface enabling beam steering applications (see section “Phase-change gradient-metasurface for dynamic beam steering” and Fig. 4 in the revised manuscript)**. We only considered two extreme states of GST, i.e., amorphous and fully crystalline, to realize a switchable optical functionality which is beam deflection to the +1st and 0th orders, respectively. Here, the change in the diameter of the nanodisk leads to a large spectral shift of the SPP resonance. As a result, a phase span as large as 315° can be achieved. For each nanodisk in the array represented in Fig. 4, we have now added the corresponding reflection amplitude and phase. It is calculated that high average reflectance over 26% and 48% can be accessed for A-GST and C-GST cases, respectively, at the expense of losing local addressability.

With this clarification, we would like to mention that due to the lossy nature of both phase-change and plasmonic materials, it is quite challenging to ideally realize the Huygens’ surfaces concept [*Advanced optical materials* 3.6 (2015): 813-820], in which magnetic and electric dipolar resonances of equal strengths are spectrally overlapped leading to high optical performance in

terms of both reflection amplitude and phase. To mitigate this issue, more complex meta-atom geometries such as split-ring resonators [*Physical review letters* 95.20 (2005): 203901], multilayered metasurfaces [*Nano letters* 14.5 (2014): 2491-2497], or freeform structures [*Nature Photonics* 15.2 (2021): 77-90], with strong electric and magnetic polarizabilities (for example through using circular displacement currents) are necessary. We also recently demonstrated the potential of metallo-dielectric meta-atoms with reduced dimensions for selective control of the hybrid plasmonic-photonic resonances of the metasurface via dynamic change of optical constants of GST without compromising the scattering efficiency [*Nano letters* 21.3 (2021): 1238-1245]. Geometric metasurfaces implementing Panchartnam-Berry phase through rotating birefringent elements can be another alternative at the expense of limiting the polarization state to the circular polarization [*Nature Nanotechnology* 10.4 (2015): 308-312]. Finally, it is notable that though we leverage a plasmonic metasurface to enhance the light-matter interaction within an ultrathin GST layer, our calculated average reflectance (for a phase span of 180°) surpasses the theoretical efficiency of several important experimental demonstrations such as [*Nano letters* 16.9 (2016): 5319-5325] with less than 5%, [*ACS nano* 14.6 (2020): 6912-6920] with less than 15%, and [*Nano letters* 19.6 (2019): 3961-3968] with less than 10% average reflectance, to name a few.

Reviewer #2 (Remarks to the Author):

Abdollahramezani and his co-workers present a phase-change material (PCM) based meta-reflector, that is electrically tunable, allowing around 10-fold reflectance change in the infrared spectral range around 1.5 microns. The device is based on a micro-heating platform to reproducibly control the phase transition of a GST layer, on top of which plasmonic nanostructures are placed. The authors claim that they reach record high reflectance of ~80%, tunable by a factor of ~10 with very high switching speed and the capability to controllably address intermediate switching states. The paper is well illustrated and the results are mostly convincing (with the exception of the last figure, see below). However, I find the structure of the manuscript not too clear, interesting and important analysis of the physics is out-sourced to the supplementary pdf and the discussion remain often quite superficial. My main concern is however the novelty over several recently published, similar results, which should be presented more clearly. These and other points that should be addressed are detailed below. I recommend to either perform a major revision to render the paper impactful enough for a broad-audience high impact journal like Nat. Comms. Alternatively the authors could submit their work to a less ambitious journal upon minor revisions of the unclear parts.

Our response: We thank the Reviewer for the careful consideration of the manuscript and the insightful questions. We are sorry that some ambiguities in the structure of the manuscript gave rise to those feelings about the novelty, clarity, and interestingness of our work. To convey a more coherent message, we reformed some parts in the revised version and added several discussion sections in Supplementary Information. Though the necessary analysis on the physics of underlying modes is embedded in the main text, to meet the specific needs of the broad audiences of *Nature Communications*, we decided to move the detailed exploration to Methods in the manuscript. We believe there is a misunderstanding with the novelty of our work over the existing

demonstrations which needs in detailed clarification (see below). **We have carried out a series of new calculations and experiments following the Reviewer's concerns to complement our study by demonstrating beam steering applications.** Thanks to the raised constructive suggestions/comments, we believe the quality of our manuscript is now significantly improved.

- My main concern is the novelty, mostly with respect to references [31,32], since the platform seems quite similar. The authors mainly seem to state that their reflectance is higher, due to less mechanical deformation and a somehow varied material stack. I feel that this is rather little novelty for a high impact journal such as Nat. Comms. I feel that for example an experimental demo of the deflection switching device, simulated in the SI Fig S6 would represent an appealing novelty.

Our response: Though our work shares a similar functionality with the two recently published works in *Nature Nanotechnology* (i.e., amplitude modulation), we believe that the novel device design and approach introduced here are important to address major concerns with the existing demonstrations and open new directions in reversible multistate phase-change metasurfaces. We would like to highlight the major aspects of our work, which surpass the existing works in the references mentioned by the Reviewer in different dimensions. The record high average reflectance of ~ 80% and optical contrast by more than eleven-fold are achieved in our work by using a new heterostructure platform. This platform is comprised of an optically smart hybrid metasurface decoupled from a stand-alone microheater of a refractory metal, in contrast to using plasmonic metals with low melting temperature used in Ref. [32] (Ref. [33] in the revised manuscript). In addition to the completely reversible switching between the amorphous and crystalline states, we show for the first time the realization of multiple nonvolatile intermediate GST states in a repeatable fashion. The extremely thin nature of the employed GST film, with highest index contrast among all PCMs, enables the realization of a repeatable and reliable amorphization process while avoiding elemental segregation as a major failure mechanism of thick GSST elements used in Ref. [31] (Ref. [32] in the revised manuscript). Furthermore, in contrast to Ref. [32], quasi-continuous spectral tuning over 250 nm with high scattering efficiency is achieved in our work in the near-infrared spectral range thanks to the engineered modal overlap with the GST layer in its different structural states. Finally, our platform enables three orders-of-magnitude faster registration of the crystalline state with lower dynamic power in GST in comparison to the other platforms using slow PCMs such as GSST (e.g., Ref. [31]). We strongly believe that these unique features provide non-incremental performance improvements over Refs. [31] and [32] as well as all other related reports. We did our best to convey a coherent signal on these important aspects in the last paragraph of Introduction in the revised manuscript.

Following the Reviewer's suggestion, we performed extra experiments to demonstrate, for the first time to our knowledge, an electrically tunable GST-based metasurface for beam forming applications. We have added the following section to the main text. We hope the added results and the explanations above address the Reviewer's concern about the novelty of our paper.

“Phase-change gradient-metasurface for dynamic beam steering. In addition to the active control over the amplitude and resonance features of the reflection spectrum, the studied hybrid platform offers tuning over the phase response of the scattered field enabling dynamic wavefront

engineering in the near-IR spectral range. Here, we demonstrate a reconfigurable phase-change gradient metasurface, called meta-deflector, to selectively steer the incident light to the +1st/0th diffracted order when GST is in its amorphous/crystalline state.

The meta-deflector is formed by a 2D array of supercells each consisting of a linear arrangement of 7 meta-atoms (as shown in Fig. 4a) following the working principle of phase-gradient metasurfaces. By monolithically increasing the diameter of Au nanodisks, the SR-SPP mode supported by the metasurface with A-GST undergoes a gradual spectral shift, and thereby $\sim 45^\circ$ phase shift is progressively added per meta-atom to the reflected wavefront (see Fig. 4a(i)). The more than 315° optical phase coverage achieved upon this evolution is enough for several phase-based optical functionalities, though this comes at the cost of variation in the reflectance on account of the on-resonance operation of the meta-atoms at $\lambda = 1495$ nm (see Fig. 4a(ii)). As a consequence of full crystallization, all designed meta-atoms serve as similar phase retarders, with $\sim 0^\circ$ phase shift between neighboring meta-atoms, providing uniform scattering efficiency alongside the supercell (see Fig. 4b(i)). The rather flat nature of the reflection amplitude and phase curves in Fig. 4b(ii) stems from the off-resonance characteristics of the LR-SPP mode at the operational wavelength.

Following the generalized Snell's law, we design a meta-deflector such that it enables the steering of light beam to the angles of 20° and 0° in the case of A-GST and C-GST, respectively. The simulated deflection intensities as a function of the outgoing angle, displayed in 2D maps in Figs. 4c(i) and 4c(ii), are in good agreement with the calculated angles from the theory. The switching contrast ratio, defined as $I_{A-GST}^{20^\circ} I_{C-GST}^{0^\circ} / I_{A-GST}^{0^\circ} I_{C-GST}^{20^\circ}$, where I represents the reflection intensity, is calculated to be 10.8 dB, which is comparable to state-of-the-art reconfigurable beam-steering metasurfaces [4, 5]. A rather non-dispersive behavior is also observed in the angular reflection responses for the spectral range of 30 nm around the designed wavelength (i.e., $\lambda = 1495$ nm) that facilitates the device operation in the S band. The steering capability of the meta-deflector is experimentally validated by fabrication of a $100 \mu\text{m} \times 100 \mu\text{m}$ phase-change metasurface (see Fig. 4d) electrically tunable using the Joule heating process (see Supplementary Information Note 1 and Fig. S9 for details). The far-field radiation patterns for 3 different wavelengths presented in Fig. 4e justify anomalous to specular reflection operation upon switching the state of GST from amorphous to crystalline. It should be mentioned that in the amorphous case, although under normal illumination the deflected beam is mainly coupled to the +1st order, still the 0th order carries noticeable portion of the reflected energy. We anticipate that the adoption of an enlarged-aperture meta-deflector comprising optimized meta-atoms with free-form geometries coupled to different states of GST can further improve the optical performance of these phase-change gradient-metasurfaces for on-demand beam forming applications such as varifocal lensing.”

Figure R3: Demonstration of an electrically driven dynamic phase-change meta-deflector. (a, b) Wavefront evolution from a supercell of the meta-deflector with (a) A-GST and (b) C-GST. (i) Simulated scattered magnetic fields from 7 constitutive meta-atoms (with diameters d_{Au} of 120 nm, 196 nm, 206 nm, 212 nm, 216 nm, 224 nm and 270 nm) of the supercell at the same time instant. The phase shift increment induced by neighboring meta-atoms can shift the position of the peak of the field up to a wavelength (as shown by the black dashed line) in the amorphous state. Each strip is calculated by an independent full-wave numerical simulation of a meta-atom illuminated by a x-polarized plane-wave with free-space wavelength $\lambda = 1495$ nm. The scale bar in the inset is 400 nm. (ii) Reflectance (left axis) and phase shift (right axis) of the scattered field from the meta-atom as a function of the Au nanodisk diameter. Discrete amplitude and phase values associated with each meta-atom in (i) are defined on the curves. (c) Simulated normalized deflection intensities from the dynamic meta-deflector as a function of the steering angle and wavelength for the cases of (i) A-GST and (ii) C-GST. (d) Angled SEM image of the fabricated sample with false-colored Au nanodisks in a supercell defined by the dashed box. (e) Measured normalized far-field radiation intensity of the active meta-deflector upon conversion of the structural phase of GST from (i) A-GST to (ii) C-GST using the Joule heating process. Images are obtained from the CCD camera in Fig. 2b for three incident wavelengths.

- How does the presented platform compare to more traditional tunable reflectance devices such as liquid crystal based approaches?

Our response: Generally, liquid crystals (LCs) comprising of uniaxial birefringent molecules can possess different switchable material phases (characterized by the orientation or order of their molecules) upon applying an electrical or thermal stimulus. For instance, commonly used nematic LCs exhibit an ordinary refractive index of 1.51 and an extraordinary refractive index of 1.70 at visible wavelengths [*Applied Physics Letters* 110.7 (2017): 071109]. Electrically switchable LC-based metasurfaces are integrated in a compact cell with two electrodes on top and bottom to facilitate the formation of a uniform electric field for reorienting the birefringent molecules. By

applying an electric field perpendicular to the surface of the substrate, the initially parallelized molecule (with respect to the substrate) can be aligned to the electric field with an in-plane refractive index change of 0.19. In thermally actuated LC-based metasurfaces, the LC cell is heated above the phase transition temperature to break down the alignment of the molecules in the nematic phase resulting in an average refractive index change of 0.15 [*ACS Photonics* 5.5 (2018): 1742-1748]. LCs are promising material platforms to facilitate the tuning of optical functionalities in a wide range of spectrum from visible to the microwave regime. In addition, the technological advances in LC-based displays and spatial light modulators have promoted their potentials for reconfigurable flat optical systems. Nevertheless, there are some major drawbacks with LC-actuated metasurfaces that have hindered the realization of high-performance adaptive meta-optics. The refractive index change between two arbitrary phases of LCs is quite smaller than the index contrast between the amorphous and crystalline states of GST (e.g., ~ 0.19 in comparison to ~ 2.7 in the near-infrared range). In addition, LCs feature volatile characteristics resulting in non-zero dynamic power consumption and bistable properties with no intermediate states that hampers the realization of continuously tunable optical functionalities. Also, local addressing of individual meta-atoms in a LC-based metasurface is quite challenging, if not impossible, since the refractive index change occurs globally across the bulky LC layer [*Science* 364.6445 (2019): 1087-1090]. Furthermore, LC cells are optically thick, ranging between 2-5 μm that downgrade the subwavelength thick nature of flat optical metasurfaces. In addition, the anchoring effect negatively affects the correct alignment of the LC molecules to the near-field of the meta-atoms, which degrades the optical performance of the device [*Advanced Optical Materials* 3.5 (2015): 674-679]. LC cells also pose different changes to different incident polarization states, which inherently restricts the operation of metasurfaces to a certain linear polarization. Accordingly, new fabrication strategies as well as ingenious design paradigms of metasurfaces would be the core of future research to enable practical LC-based metasurfaces for real-world applications. An overview of the existing adaptive metasurfaces based on their key performance measures is detailed in the table below.

Table R1. Comparison of different reconfiguration methods for adaptive metasurfaces.

Method	Material	Stimulus	Modulation	Speed	Energy	Wavelength range	Functionality	
phase transition	LC	thermal [1]	amplitude (~5-fold)	NA	NA	1.55 μm	switching	
		electrical [2]	amplitude (~5-fold)	NA	NA	1.55 μm	switching	
		electrical [3]	amplitude	NA	NA	VIS	coloration	
		electrical [4]	amplitude	NA	NA	VIS	beam deflection	
	VO ₂	electrical [5]	amplitude (~%85)	~1 s	~1 μJ	3-4 μm	imaging	
		electrical [6]	amplitude (~%33)	1.27 ms	~21 nJ	1.1 μm	switching	
		thermal [7]	amplitude (~%80)	NA	NA	11.6 μm	switching	
		thermal [8]	amplitude	NA	NA	1.5-5 μm	switching	
		optical [9]	amplitude (~4-fold)	40 ms	11.5 mW	10.6 μm	holographic imaging	
		GLS	electrical [10]	amplitude (15%)	10 ms	NA	1.55 μm	switching
		GST	optical [11]	phase	phase	200 ns	6 mW	1.55 μm
	optical [12]		amplitude	amplitude	85 fs	9 μJ	730 nm	beam focusing
	thermal [13]		emission	emission	NA	NA	6.5-9.5 μm	beam emitting
	thermal [14]		polarization	polarization	NA	NA	3.1 μm	beam focusing
	electrical [15]		amplitude	amplitude	100 ns	100 μJ	350-750 nm	holographic imaging
	optical [16]		phase	phase	NA	4 mW	1.55 μm	beam focusing
	thermal [17]		emission	emission	NA	NA	3.4-3.9 μm	thermal imaging
carrier doping	graphene	electrical [18]	phase	NA	40 V	7.7 μm	polarizing	
		electrical [19]	amplitude	NA	60 V	1.8 μm	switching	
		electrical [20]	amplitude (47%)	100 kHz	350 V	300 μm	switching	
		electrical [21]	phase	NA	170 V	8.7 μm	switching	
	ITO	electrical [22]	amplitude	amplitude	500 kHz	20 V	4 μm	switching
		electrical [23]	phase (π)	phase (π)	500 kHz	2.5 V	1.55 μm	beam steering
		electrical [24]	phase (π)	phase (π)	NA	80 V	6 μm	polarizing
	InSb	thermal [25]	phase ($3\pi/2$)	phase ($3\pi/2$)	NA	NA	11.7 μm	beam steering
	Perovskite	thermal [25]	amplitude (100%)	amplitude (100%)	500 ps	35 μJ	1.55 μm	switching
elasticity	PDMS	mechanical [26]	phase (2π)	phase (2π)	NA	NA	915 nm	beam focusing
		mechanical [27]	phase (2π)	phase (2π)	NA	NA	632.8 nm	beam focusing
		mechanical [28]	frequency	frequency	NA	NA	3.37 μm	beam focusing
MEMS	Si/SiN _x	electro-mechanical [29]	phase	phase	4 kHz	85 V	622-784 nm	beam focusing
	Au/SiO ₂	electro-mechanical [30]	phase	phase	1 kHz	60 V	4.6 μm	beam focusing
	Au/SiO ₂	electro-mechanical [31]	amplitude (56%)	amplitude (56%)	30 kHz	16 V	6.3 μm	switching
Lorentz force		electrical [32]	amplitude (2.5%)	amplitude (2.5%)	5 μs	NA	1 μm	switching

- [1] ACS nano 9(4), 4308–4315 (2015).
 [2] Optics express 21(2), 1633–1638 (2013).
 [3] Nature communications 6, 7337 (2015).
 [4] Optics express 24(15), 16815–16821 (2016).
 [5] Nature communications 7, 13236 (2016).
 [6] Nano letters 17(8), 4881–4885 (2017).
 [7] Applied Physics Letters 101(22), 221101 (2012).
 [8] Optics express 17(20), 18330–18339 (2009).
 [9] Advanced Materials 30(5) (2018).
 [10] Applied Physics Letters 96(14), 143105 (2010).
 [11] Advanced Functional Materials 28.10 (2018): 1704993
 [12] Nature Photonics 10(1), 60 (2016).
 [13] Laser & Photonics Reviews 11(5) (2017).
 [14] Light: Science & Applications 6(7), e17016 (2017).
 [15] Nature 511(7508), 206 (2014).
 [16] Scientific reports 5, 8660 (2015).
 [17] Advanced Materials 27(31), 4597–4603 (2015).
 [18] Nano letters 14(11), 6526–6532 (2014).
 [19] Nano letters 14(1), 78–82 (2013).
 [20] Nature materials 11(11), 936 (2012).
 [21] Nano letters 17(5), 3027–3034 (2017).
 [22] Applied Physics Letters 102(22), 221102 (2013).
 [23] Nano letters 16(9), 5319–5325 (2016).
 [24] Nano letters 17(1), 407–413 (2016).
 [25] Nature communications 8(1), 472 (2017).
 [26] Laser & Photonics Reviews 10(6), 1002–1008 (2016).
 [27] Nano letters 16(4), 2818–2823 (2016).
 [28] Nano letters 10(10), 4222–4227 (2010).
 [29] Nature communications 9(1), 812 (2018).
 [30] APL Photonics 3(2), 021302 (2018).
 [31] Advanced Optical Materials 1(8), 559–562 (2013).
 [32] Nature communications 6, 7021 (2015).

- It seems that the platform does not allow control over a full 2π phase cycle, in the SI 180 degrees are demonstrated, but maybe this can be combined by hybridization with some localized surface plasmon or so to go beyond 1π . Can the authors comment on that?

Our response: We would like to clarify that we have considered two different strategies for realization of gradient phase modulation: i) local tuning of the GST cell in each individual meta-atom (Supplementary Fig. S10), and ii) global control over the whole GST film embedded in the metasurface (Fig. 4 in the revised manuscript). For the former case, our criterion is to design a meta-atom that can cover 180° phase shift with minimum reflectance variation upon transition of GST from the amorphous state all the way to the fully crystalline state with 20% crystallization steps (see Supplementary Fig. S10). This enables realization of continuously tunable optical functionalities, e.g., beam steering, holographic imaging, and so on. Since there exists a single SPP mode (pure or hybrid) at each crystallization level governing the optical performance of the metasurface, the optical efficiency is fundamentally restricted due to a strong correlation between the scattering amplitude and phase. On the latter case, **we have dedicated extra efforts to experimentally demonstrate, for the first time to our knowledge, an electrically tunable GST-based metasurface enabling beam steering applications (see section “Phase-change gradient-metasurface for dynamic beam steering” and Fig. 4 in the revised manuscript)**. We only considered two extreme states of GST, i.e., amorphous and fully crystalline, to realize a switchable optical functionality which is beam deflection to the +1st and 0th orders, respectively. Here, the change in the diameter of the nanodisk leads to a large spectral shift of the SPP resonance. As a result, a phase span as large as 315° can be achieved. For each nanodisk in the array represented in Fig. 4, we have now added the corresponding reflection amplitude and phase. It is calculated that high average reflectance over 26% and 48% can be accessed for A-GST and C-GST cases, respectively, at the expense of losing local addressability.

With this clarification, we would like to mention that due to the lossy nature of both phase-change and plasmonic materials, it is quite challenging to ideally realize the Huygens’ surfaces concept [*Advanced optical materials* 3.6 (2015): 813-820], in which magnetic and electric dipolar resonances of equal strengths are spectrally overlapped leading to high optical performance in terms of both reflection amplitude and phase. To mitigate this issue, more complex meta-atom geometries such as split-ring resonators [*Physical review letters* 95.20 (2005): 203901], multilayered metasurfaces [*Nano letters* 14.5 (2014): 2491-2497], or freeform structures [*Nature Photonics* 15.2 (2021): 77-90], with strong electric and magnetic polarizabilities (for example through using circular displacement currents) are necessary. We also recently demonstrated the potential of metallo-dielectric meta-atoms with reduced dimensions for selective control of the hybrid plasmonic-photonic resonances of the metasurface via dynamic change of optical constants of GST without compromising the scattering efficiency [*Nano letters* 21.3 (2021): 1238-1245]. Geometric metasurfaces implementing Pancharthnam-Berry phase through rotating birefringent elements can be another alternative at the expense of limiting the polarization state to the circular polarization [*Nature Nanotechnology* 10.4 (2015): 308-312]. Finally, it is notable that though we leverage a plasmonic metasurface to enhance the light-matter interaction within an ultrathin GST layer, our calculated average reflectance (for a phase span of 180°) surpasses the theoretical efficiency of several important experimental demonstrations such as [*Nano letters* 16.9 (2016):

5319-5325] with less than 5%, [ACS nano 14.6 (2020): 6912-6920] with less than 15%, and [Nano letters 19.6 (2019): 3961-3968] with less than 10% average reflectance, to name a few.

- In the introduction, the authors state that their platform is the first fully reversible reconfigurable GST-based metasurface. To my knowledge other works claim to achieve also fully reversible GST metasurfaces [e.g. Julian et al, Optica 7(7) 746-754, 2020], but also [31] and [32]. the authors should lower their claims accordingly.

Our response: We would like to first restate the full sentence of our claim in the manuscript: “Being the first demonstration of a fully reversible reconfigurable GST-based metasurface with multiple intermediate states and a large tuning range, our platform has the potential for major applications in several fields including imaging, computing, and sensing.”. Regarding the Reviewer’s question, it should be emphasized that Refs. [31, 32] (Refs. [32, 33] in the revised version) are the only experimental demonstrations of electrically actuated phase-change metasurfaces in the literature. While the functional material in Ref. [31] is GSST (featuring slower registration of the crystalline state with a higher dynamic power and a lower refractive index contrast compared to GST), Ref. [32] failed to demonstrate multi-state operation in the GST-based metasurface. Other works in the field [e.g., Julian et al, Optica 7(7) 746-754, 2020] leveraged ultrafast laser pulses for the reamorphization process, which hinders full integration of the device with the existing optoelectronic circuits. We are sorry that “electrically driven” was dropped from the above sentence that caused such sense of overclaiming. **We have rephrased the sentence as follows: “Being the first demonstration of an electrically driven, fully reversible reconfigurable GST-based metasurface with multiple intermediate states and a large tuning range,”.** We have compared the state-of-the-art GST-based metasurfaces using non-electrical stimulus in the above table.

- What are the differences of the here presented micro-heaters in comparison to what is commercially developed for phase-change RAM applications?

Our response: Generally, each element of phase-change random access memories (PCRAMs) is formed through vertical integration of a top electrode contact (TEC) and a PCM cell that is directly connected to a small-scale thermally stable and chemically inert metal heater in contact with the bottom electrode contact (BEC). The BEC’s access to the PCM cell is forbidden through integration of an insulator spacer incorporating the heater. Depending on the interface size of the metal heater and the PCM cell, these structures fall into one of two general categories [*Journal of Vacuum Science & Technology B, Nanotechnology and Microelectronics: Materials, Processing, Measurement, and Phenomena* 28.2 (2010): 223-262, *Proceedings of the IEEE* 98.12 (2010): 2201-2227]: i) contact-minimized cell structures where a narrow cylindrical heater contacts a thin film of PCM, and ii) volume-minimized cells where the volume of the PCM element is confined to have a small cross section with the heater. The most common design of the former case is the so-called mushroom cell, which owes its name to the shape of the programming area in the PCM cell. Other cell designs under these two categories also exist in the literature including edge-contact, μ trench, ring, pillar, and lance structures [*Phase change materials. Springer, Boston, MA, 2009. 355-380*]. In PCRAMS, amorphization of the programming region is performed through a high-current pulse with a short trailing edge delivered to the heater to rapidly melt the top PCM

layer followed by a quenching process that freezes the material state to the disordered form. On the other hand, crystallization is carried out via applying a longer pulse with lower intensity to heat the programming region above the glass transition temperature for a long enough time to induce ordered transition. Apart from the more established vertical structures, less-integrated in-plane devices such as the bridge and line structures where the resistance of the PCM cell itself is exploited to produce the necessary heat for the conversion process were also pursued [*Nature materials* 4.4 (2005): 347-352, 2006 *International Electron Devices Meeting, 2006*, pp. 1-4, DOI: 10.1109/IEDM.2006.346910].

While the amorphization/crystallization mechanisms through the Joule heating process is to some extent similar in both PCRAMs and phase-change nanophotonics, overall, exploiting the heater configurations used in PCRAMs places stringent constraints on the targeted optical performance of PCM-based metasurfaces. First, to adapt to the increasing number of memory cells per chip and to alleviate the impact of the resistance variation and charge leakage, the physical dimensions of PCRAM heaters need to be shrinking. The miniaturized scale of each cell fairly facilitates the quenching process as the small programmable region of the PCM element is in direct contact of two metallic parts with high thermal conductivity. This is not the case for phase-change metasurfaces with large aperture sizes where successful uniform heat generation and melt-quenching processes must be carried out simultaneously using two-orders of magnitude larger microheaters. Second, PCRAMs leverage a crossbar architecture of metallic bit-lines/word-lines to address vertically integrated PCM elements whereas integration of metallic circuitries with optical metasurfaces imposes significant dissipative loss and degrades the overall efficiency. Third, formation of an arbitrary crystallization filamentation as a direct current path through the PCM element in the in-plane PCRAMs prevents uniform phase transition of the whole volume of PCM in large-scale metasurfaces. To address these issues with PCRAM heater, our large-scale heterostructure architecture capitalizes on the integration of a robust resistive microheater that i) is decoupled from the PCM layer to sustain high scattering efficiency due to the excessive dissipative loss of the tungsten microheater, ii) facilitates the quenching process on account of wise selection of stacked materials embedded the PCM layer, and iii) enables uniform heat generation for successful transition from the amorphous state to multiple intermediate states. The successful operation of our electrically driven reprogrammable metasurface owes to a judicious co-optimization of a multiphysics model taking into account the extreme electrical, thermal, and optical properties of the contributing materials.

- Would the approach work also in a pure layer-stack geometry (thus without the plasmonic nano-discs on top), similar to recently reported radiative cooling applications [e.g. Raman et al. *Nature* 515, 540–544 (2014)]?

Our response: The paper shared by the Reviewer reports an interesting approach to utilize an engineered multilayer stack of dielectric media on top of a metallic back-reflector to enhance reflection in the visible/near-IR spectral range and simultaneously increase absorptivity at the mid-IR window. Though on-demand reflectors/absorbers can be designed using passive one-dimensional photonic crystal slabs, tuning their optical performances at will by adding active PCM slabs faces some challenges from both experimental and fundamental points of view. Since

elemental segregation upon repeated switching is a typical failure mechanism of thick PCM cells, remixing of elements through complete melting of the PCM after each reset pulse is crucial. This necessitates application of an ultrathin PCM layer to guarantee repeatable and reliable switching operations. In addition, to realize a successful quenching process, materials with high thermal conductivity must be used as the surrounding media of the PCM layer. This requires exclusion of most thick dielectric layers featuring low thermal conductivity from the design library. Furthermore, within the stacked layers, the position of the PCM layer should be considered in proximity of the substrate that functions as a good heat sink. From the design point of view, although the low-loss feature of amorphous GST facilitates design of multi-layered structures relying on Fabry–Perot-type constructive/destructive interferences, the highly dispersive and absorbing crystalline GST violates this paradigm. To better understand the effect of the plasmonic metasurface on the overall optical performance of the device, we have simulated the heterostructure without considering the array of nanodisks. The figure below shows the reflectance spectra for different crystallization levels of GST with step of 20%. It is evident that the maximum reflectance contrast between the A-GST (i.e., 0% crystallization) and C-GST (i.e., 100% crystallization) is $\sim 30\%$, which is quite lower than $\sim 80\%$ calculated for the maximum reflectivity of the metasurface device. The multilevel tunability with high modulation depth granted by our phase-change metasurface stems from enhanced light-matter interaction due to the excitation of surface plasmon polaritons coupled to the available state of an ultrathin layer of GST to facilitate manipulation of reflectance.

Figure R4: Multi-state operation of a tunable multilayered stack incorporating a phase-change film for controlling the reflectance in the near-IR regime (inspired by [*Nature* 515.7528 (2014): 540-544]). Inset: electric field magnitude distributions for the A-GST (i.e., 0% crystallization) and C-GST (i.e., 100% crystallization) cases.

- The usefulness of the section "Performance analysis using machine learning" is imo quite doubtful. In this context the authors state that the dimensionality reduction is "preserving necessary information (e.g., nature of the governing mode)", which falsely might sound as if the unsupervised learning would automatically extract physical information. This is however not the case, it just clusters the data into some kind of principal components, regardless the underlying mechanism. Linking the latent space to physics can be actually highly non-trivial, the discussion here can be misleading in this regard. It can also mislead the reader to state that the fact that A- and C-GST points span the full 3D t-SNE latent space has some physical meaning about a broad coverage of

different possible reflection responses through the tested designs. t-SNE is actually built such that it tries to distribute clusters over the full target latent space. Using for instance only c-GST would probably also lead to a more or less fully covered latent-space. The authors might look into [Kiarashinejad et al. *Advanced Intelligent Systems* 2, 1900132, 2020] for an example how to use dimensionality reduction to quantitatively assess the feasible physical responses of an optical system.

Our response: We thank the Reviewer for this comment and for referring to our previous work [Kiarashinejad et al. *Advanced Intelligent Systems* 2, 1900132, 2020]. Thanks to the fascinating cognitive capabilities of humans through visual perception, data visualization techniques for representation of datasets governed by complex relations has been always a useful tool. The authors absolutely agree with the Reviewer that neither linear nor nonlinear dimensionality reduction techniques can “automatically extract physical information”. However, we think this conclusion is not implied by “preserving necessary information” notion, which has been explicitly noted and proven as the basis of t-SNE, as a state-of-the-art non-parametric, nonlinear dimensionality reduction technique with the major purpose of data visualization [*JMLR* 9.86, 2579-2605 (2008), *Data Mining and Knowledge Discovery* 5.2 (2015): 51-73, *Proceedings of the IEEE* 98.6 (2010): 959-971, *IEEE transactions on geoscience and remote sensing* 44.6 (2006): 1586-1600]. While we firmly believe that unveiling the nature of the governing modes through the inspection of t-SNE plots is not straightforward, if not impossible, the electromagnetic modes that governs the spectral responses of the metasurface (or equivalently shape the local and global structure of datapoints in the high-dimensional space) is an important hidden information that is likely preserved during the transformation. **Nevertheless, to prevent any misleading, we have added the following sentence to the main text: “This should not be translated as the ability of data visualization techniques to reveal underlying physics of light-matter interaction or extract physical information.”.**

We concur with the Reviewer that there exists no direct link between the results of the spatially extended clusters in the latent space and the “physics” of the problem. The essence of using t-SNE is to trustfully represent complex datasets with intrinsically high dimensions in low-dimensional spaces while preserving as much relevant information as possible [*Neural Computation* 24.3 (2012): 771-804, *Distill* 1.10 (2016): e2]. However, t-SNE can uncover hidden structures in the high-dimensional dataset through capturing much of the local structures, while also revealing global structures such as the presence of clusters [*Journal of machine learning research* 9.11 (2008): 2579-2605, *Nature biotechnology* 37.1 (2019): 38-44]. This is possible mainly due to two unique features of t-SNE. First, this algorithm seeks to preserve the pairwise distances between data points by translating Euclidean distances between data points in the high-dimensional space into conditional probabilities that represent “similarities” [*Journal of Machine Learning Research* 15.1 (2014): 3221-3245]. In projection of the high-dimensional data to the low-dimensional space, t-SNE tends to position the points on a plane (or hyperplane) such that the pairwise distances minimize a cost function, which is a measure of the similarity between two probability distributions [*Distill* 1.10 (2016): e2]. In addition, t-SNE fairly ameliorate the “crowding problem” that is distinguished by the overlap of projected datapoints in the center of the low-dimensional map due to the excessive attractive forces between moderately distant datapoints in the original space [*Journal of machine learning research* 9.11 (2008): 2579-2605]. In this regard, distinctly

isolated clusters of similar datapoints sharing multiple features can be readily identified in the low-dimensional space, though t-SNE is not a dedicated clustering algorithm [*Nature methods* 16.3 (2019): 243-245, *Nature communications* 10.1 (2019): 1-14, *SIAM Journal on Mathematics of Data Science* 1.2 (2019): 313-332]. In our case, the algorithm forms two widely separated natural clusters of similar points, i.e., for A-GST and C-GST cases (blue and red dots in Fig. 4a). The minimum overlap between two clusters implies that despite the variation of the structural parameters, each state of GST can only provide a specific cluster of datapoints, which are different in nature from those in the other cluster. We think that the wording in the original manuscript did not convey the main message, which is the emphasis on “two widely unfolded clusters” and not “span of the full 3D t-SNE latent space”. Noteworthy, the axes of a t-SNE plot are abstract scores describing complex curved paths in the original space and are not meant to be straightforwardly interpretable in terms of the axis/units of the original high-dimensional space. This is the reason why t-SNE plots without scaling are also common in the literature. In addition, unlike simple linear dimensionality-reduction algorithms such as principal component analysis (PCA) whose plot axes are weighted linear combinations of the original dimensions, as specified by the principal eigenvectors of the covariance matrix, the axes in t-SNE plots do not convey any specific information [*Distill* 1.10 (2016): e2]. **Having said that, to avoid any misinterpretation, we have rephrased the corresponding paragraph in the revised manuscript as follows: “The implication of the formation of two distinguishable unfolded clusters corresponding to A-GST (blue points) and C-GST (red points) in the 3D latent space is twofold. Incorporation of GST in the meta-atom grants a different class of responses not easily accessible through just variation of structural parameters with one GST state. Accordingly, the GST crystallization state can be considered as an effective tuning knob to modify the metasurface performance. Furthermore, the metasurfaces with A-GST and C-GST are likely governed by modes with distinct natures.”.**

To complete our discussion, we investigate the evolution of electric field distribution for three simulated metasurfaces with different structural parameters for both A-GST and C-GST cases (see Supplementary Fig. S11a, also shown below). For each state of GST, comparable mode profiles can be observed for all 3 randomly selected samples. While the top row illustrates the excitation of short-range surface plasmon mode at the interface of the gold nanodisk and the alumina layer, the field enhancement in the bottom row mainly occurs at the tips of the gold nanodisk and interface of the gold back-reflector due to the excitation of long-range surface plasmon polaritons. The reduced-dimensional responses are indicated in the 3D latent space in Supplementary Fig. S11b using color-coded shapes. While this should not be interpreted as the “physics extraction capability”, at least it gives a tangible insight on the linking between some randomly selected datapoints within one cluster to the mode profile of the corresponding meta-atoms.

Figure R5: (a) Inspection of the normalized electric field magnitude at the resonance wavelengths of the meta-switch with A-GST (top row) and C-GST (bottom row) in the $x-z$ plane of a meta-atom. The structural parameters of the studied metasurface are $p = 440$ nm, $d_{Au} = 210$ nm, and $t_{GST} = 25$ nm (first column), $p = 580$ nm, $d_{Au} = 190$ nm, $t_{GST} = 35$ nm (second column), and $p = 750$ nm, $d_{Au} = 230$ nm, $t_{GST} = 45$ nm (third column). (b) The embeddings corresponding to A-GST and C-GST in the latent space. The reduced-dimensional reflectance responses of metasurfaces in panel (a) are depicted using color-coded shapes.

We would like to conclude that while visual representation of high-dimensional data in a low-dimensional space is imperative for exploratory data analysis, certain attention should be paid to prevent some common misreading of t-SNE plots. 1) The relative distance between well-separated clusters in the low-dimensional space does not provide any intuition. In fact, t-SNE does not retain distances but probabilities, so Euclidean distances in high-dimensional and low-dimensional spaces do not provide any useful measure about the response of the metasurface. 2) The relative sizes of the clusters in t-SNE plots do not convey any specific meaning. This stems from the fact that t-SNE tends to expand/contracts the dense/sparse data clusters of the high-dimensional space in the projected embeddings. 3) The topology of generated clusters in the latent space does not provide any information about the nature of the dataset in the original dimension. The algorithm hyperparameters fairly affect the elongation, curvature, or clumping of the generated clusters. 4) Successive runs of the t-SNE algorithm do not necessarily generate similar outputs. This follows from the rationale that some hyperparameters like perplexity, a measure of the effective nearest neighbors, play a key role in the optimization process. Despite its multiple strengths in data visualization, t-SNE has some weaknesses. The main functionality of t-SNE is for visualization purposes so using it as a pre-processing machine-learning approach is tricky, if not improper. Also, t-SNE has a low performance on datasets with high intrinsic dimensions especially given the nonconvexity of the cost function that requires optimization of several hyperparameters.

Following the suggestion by the Reviewer, we have added two more quantitative analyses on assessing the feasible physical responses offered by A-GST and C-GST metasurfaces. By

leveraging a hybrid approach consisting of both dimensionality reduction and i) convex hull or ii) one-class support vector machine (SVM) techniques, the range and degree of feasible responses are studied in the latent space. We limit our discussion to the main outcomes of these approaches, as the working principles and theoretical frameworks are detailed in our previous work [Kiarashinejad *et al. Advanced Intelligent Systems* 2, 1900132, 2020]. In a nutshell, the convex hull of a dataset is the smallest convex set that includes all datapoints. Starting from a set of training data, the first approach repeatedly forms the convex hull of transformed data in the latent space upon iterative expansion of the training data fed to the dimensionality reduction algorithm. Upon convergence, the envelope of the final convex geometry can suggest a bound of the so-called feasible responses by a specific class of structures. Supplementary Fig. S12a (also shown below) displays convex hulls for A-GST and C-GST cases with blue and red, respectively, in the 3D latent space. Despite its effectiveness in visually representing binary classified data, there is a major shortcoming with this approach. The algorithm forcefully expands the convex hull to reach a convexity irrespective of the possible nonconvexity of that class. As a negative result, some datapoints from each class (so-called unfeasible) in the latent space are included in the convex hull of the other class, which degrades the effectiveness of the convex hull approach in distinguishing totally feasible responses. In addition, the convex hull approach does not provide any intuition about the level of feasibility. In the second approach, a quantitative metric is offered by one-class SVM to measure the separability of two classes (i.e., feasible and unfeasible). Basically, a one-class SVM represents a series of separating hyperplanes (corresponding to different levels of unfeasibility) that encompass all instances of a class in the latent space without any convexity restriction. The nonconvex decision boundaries constructed through the one-class SVM approach help us determine how much an object from a different class (e.g., the response of a C-GST metasurface) is close to the one class of interest (e.g., the response of an A-GST metasurface) using an outlier threshold. The graph in Supplementary Fig. S12b (also shown below) displays several contours (with different colors) surrounding the reddish geometry enclosing A-GST data shown as white dots. Those points located in color-coded counters with negative scores are predicted to be the 10% outliers for the class of A-GST. It is clear from Supplementary Fig. S12b that majority of observations laying outside the high-confidence region, characterized by positive scores, are C-GST samples (black dots), which are predicted as unfeasible responses for A-GST metasurfaces.

Figure R6: (a) Convex hull of (i) A-GST and (ii) C-GST metasurfaces representing their feasible responses in the 3D latent space. (b) Nonconvex geometry calculated by the one-class SVM algorithm for the A-GST metasurface.

The graph shows the separating hyperplanes as a measure of response feasibility in the 2D latent space. The boundary separating 10% observations as outliers from the rest of the data occurs where the contour value is 0. Black points show the responses for the C-GST case. It is clear that for the most part, they fall outside the feasible range of the A-GST metasurface. This point is also observed from panel (a).

We thank the Reviewer for raising these insightful concerns. We hope the newly added discussions in both the revised manuscript and Supplementary Note 6, Note 7, and Figs. S11-S12 will prevent possible misinterpretations of the machine-learning analysis.

- In the very last paragraph the authors mention a "wrapped algorithm". It is not clear at all to me how this algorithm works and how the authors come to the mentioned ranking of highest variability design parameters. Also, after presenting the results, the authors don't discuss the meaning of this ranking like the consequences for design optimizations or implications concerning fabrication robustness.

Our response: We appreciate the Reviewer's comment about the need for expanding and further clarifying our explanation of the working principle of the wrapper method. The wrapper method as a feature selection algorithm aims to select and rank the most significant and relevant features in a given dataset [*Artificial intelligence 97.1-2 (1997): 273-324, Journal of machine learning research 3.Mar (2003): 1157-1182*]. Beyond that, this technique helps faster training of machine learning algorithms while improves the accuracy of the model and reducing the overfitting problem. The wrapper method follows a greedy search approach by assessing all possible combinations of features against an evaluation criterion. By sequentially adding features starting from an empty feature set, the wrapper algorithm performs cross-validation by calling the evaluation criterion function for each candidate feature subset. In each sequence, the data is first divided into two subsets of training (that contains the selected subset of features and corresponding responses) and testing (that contains the complementary subset of features and corresponding

responses). By training a feedforward neural network model with the former subset, the evaluation function returns some measure of distance (or loss) of those predicted by the latter subset. Then, the returned losses are summed up and divided by the number of the test observations leading to the criterion mean, which is considered as a metric for evaluation of the rank of each feature subset. The process continues until adding more feature subsets does not decrease the mean. Here, our goal is to score each structural parameter given that the evaluation criterion satisfies maximum modulation depth at 1550 nm (with 10% bandwidth) upon switching of the state of GST from amorphous to crystalline. The selected features in order and the corresponding criterion values computed at the consecutive steps are $\{t_{\text{GST}}, d_{\text{Au}}, p, t_{\text{Al}_2\text{O}_3}\}$ and $[9.61 \times 10^{-5}, 7.30 \times 10^{-5}, 6.03 \times 10^{-5}, 5.52 \times 10^{-5}]$, respectively, as shown in Supplementary Fig. S13 (also shown below).

Figure R7: Criterion values calculated at each step of the wrapper algorithm and the corresponding features.

Note that we already discussed the effect of each structural parameter by exploring the evolution of the reflectance spectra of the metasurface in the near-IR regime (see Supplementary Note 9, Figs. S6 and S14 in the revised version). **We have added the above discussion to the Supplementary Note 8. We also completed our discussion in the manuscript as follows: “We also support these inferences through studying the evolution of underlying modes of the metasurface in the near-IR regime (see Supplementary Note 9). From the experimental point of view, the influential role of GST thickness on the modulation depth demands employing of reliable PCM growth techniques such as atomic layer deposition with high precision and uniformity.”**

- In general I find the recent works on PCM based switchable meta-reflectors ([31, 32] and the present one) really interesting and this seems to open exciting routes for further concepts. As a suggestion: I would be very interested in a discussion about the possibility to micro- or even nano-structure heating elements in order to induce a spatially heterogeneous crystallization/amorphization pattern. The first question is probably, whether this is feasible from a fabrication point of view as well as regarding the electronic engineering aspect (micro-electrode array with very dense connections to the metasurface area). The second point would be the local heating: Could it be possible to induce the phase change in a controlled way on a (sub)-micron length-scale, without affecting neighboring "phase-change cells"?

Our response: We thank the Reviewer for finding our work interesting and the constructive comments. Addressing individual meta-atoms is an important yet challenging step that can

empower dynamic multifunctional metasurfaces to prevail over conventional spatial light modulators (based on liquid crystal or microelectromechanical structures). This would be more demanding when dealing with subwavelength meta-atoms whose reconfiguration mechanism depends on Joule heating. Since stimulation of each meta-atom relies on the precise control of biasing of the laterally extended pixelated heater elements, the performance measure of the metasurface is affected by the lossy nature of the wires network. At the expense of losing addressability over one direction, this negative effect could be fairly alleviated by relying on a 1D array of resistive nanoribbons (i.e., micro-electrodes) homogeneously heating meta-atoms in the orthogonal direction. In this way, gate biasing can be applied to the end faces of individual micro-electrodes through compact contact pads without interfering with the optical beam. Along this direction, recently, a simple prototype of tunable gated field-effect metasurfaces consisting of 96 independently addressable elements with pitch size of 400 nm has been demonstrated [*ACS nano* 14.6 (2020): 6912-6920]. This along with the potential of robust sub-micrometer width tungsten electrodes as a high-temperature high-speed heater [*Review of Scientific Instruments* 87.2 (2016): 024904] hold the promise for realization of dense micro-electrodes externally controlled through an electric signal (e.g., a voltage provided by a printed circuit board).

To figure out the feasibility of local addressing of GST cells using Joule heating, electro-thermal simulations are carried out for a 1D array of heterostructure ribbons (600-nm-width, 10- μm -long) comprising tungsten micro-heaters and the reflective meta-atoms (see Supplementary Fig. S4 (also shown below)). We study the impact of the reset pulse on the phase transition of GST by applying a 200 ns-long 0.18 V electrical pulses to the micro-electrode. Supplementary Fig. S4b shows the temperature profile in a cross section at the center of the array and at the end of the reset pulse. It is evident that the center pixel can be homogeneously heated up to 630 $^{\circ}\text{C}$ that ensures complete re-amorphization. Supplementary Fig. S4b also shows that during the entire heating process, the temperature of side GST cells remains well below the onset crystallization temperature (i.e., 160 $^{\circ}\text{C}$). Such a negligible thermal crosstalk between the neighboring elements guarantees the successful addressability of the phase-change metasurface at the pixel level. **As a complementary view, we have added the above discussion to Supplementary Note 1.**

Figure R8: Simulated electrothermal model for a 1D array of heterostructure ribbons comprising tungsten micro-heaters and reflective meta-atoms. (a) Simulated temperature distribution at the cross section of the adjacent meta-atoms at the end of the reset pulse (i.e., 300 ns). (b) Transient temperature profiles in the center of the middle GST cell as well as the corner of the neighboring cells upon applying the reset pulse. While the middle cell is successfully

melted (above ~630 °C) followed by a quenching process (high cooling rate of > 1 °C/ns), the maximum temperature experienced by the neighboring cells is well below the crystallization temperature (i.e., ~160 °C).

- Personally I find the way figure 3a presents the reflectance spectra not very helpful. It would be far easier to compare the spectra quantitatively if they were plotted in a 2D plot, each spectra offset by a constant increment (similar to for example fig 4c and e).

Our response: We appreciate the Reviewer’s suggestion on presenting the reflectance spectra for different levels of crystallizations in a 2D plot. We have added the 2D format of the plots represented in Fig. 3a below. After comparing these two formats, we think that due to the available perspective view and the large spacing between the min and max values of the y axis (i.e., reflectance) in the current 3D plots, the slight differences between the experimental and simulated reflectance spectra can easier be figured out. In this regard, we would like to retain the 3D format as a more easy-to-read platform for the quantitative comparison between the reflectance spectra with different crystallization fractions.

Figure R9: 2D format of the 3D plots in Fig. 3a. Experimental (a) and simulation (b) reflectance spectra for different crystallization fractions of GST. The reflectance spectra are shifted up by one arbitrary unit for the sake of clarity.

- In the supporting information figure S7, the reproducibility is indeed clearly demonstrated, it would just be nice to plot the initial or average reflectance in every figure as a baseline for better comparison.

Our response: This is an excellent suggestion. We have revised Supplementary Fig. S7 by adding the average reflectance spectra for both A-GST and C-GST in all figures for better comparison. We have added the following text to the caption of Supplementary Fig. S7: “The average reflectance spectra for A-GST and C-GST (shown in Fig. 2c) are represented by blue and red dashed lines, respectively.”

Reviewer #3 (Remarks to the Author):

The authors demonstrate a metasurface that is dynamically and reversibly switchable between reflecting and absorbing functionalities in the NIR. They demonstrate significant performance improvements in terms of switching speed and efficiency over similar designs that have been previously reported. The results are highly significant to the field of reconfigurable optics and were

obtained via scientifically sound methodology. I believe the manuscript could be improved if the following are addressed.

Our response: We appreciate the Reviewer for the appreciation of the quality of our work with positive remarks and constructive feedback. In what follows, we address the insightful comments raised by the Reviewer.

- It is suggested that a PCM device can maintain performance for $\sim 10^{12}$ cycles, but it is not stated what is achieved for this device (only data up to 50 is presented). Even after only 50 cycles there is a noticeable upward drift in the reflectance in the C-GST state. How many cycles are achievable and what causes the limitation on this number?

Our response: Endurance limit of PCMs is a critical aspect of phase-change metasurfaces that influences the reliable operation of reconfigurable optical devices over a multitude of switching cycles. Though several degradation/failure mechanisms of PCMs have been reported in the literature, elemental segregation and void formation are the most appreciated ones [*MRS bulletin* 39.8 (2014): 703-710, *Proceedings of the IEEE* 98.12 (2010): 2201-2227]. The former stems from the electromigration of the constituent elements that negatively affects the crystallization kinetics and properties. The latter is attributed to the change in the mass density of PCMs upon transition in the structural phase, which finally leads to the formation cracks due to mechanical stresses. Besides these effects, the deviation observed in the reflectance response of C-GST after initial switching cycles is likely due to the formation of bigger nucleated islands with different fractions of two known crystalline phases of GST, i.e., face-centered-cubic and hexagonal [*Journal of Applied Physics* 97.9 (2005): 093509], and inter-diffusion of layers during repeated cycles of melt-quenching as a result of material fatigue. Similar initial cycle-to-cycle variation has been also reported in phase-change memories [*Phase Change Materials: Science and Applications* (2009): Springer, *Proceedings of the IEEE* 98.12 (2010): 2201-2227]. To ensure high reproducibility with a stabilized switching operation, an initial “conditioning” treatment is imperative, in which a single reset pulse followed by a train of power decreasing pulses are iteratively employed for a few times [*Optical Materials Express* 8.9 (2018): 2455-2470, *ACS applied materials & interfaces* 10.49 (2018): 41855-41860]. Inspired by the mature technology of phase-change memories, the following strategies can also be considered to improve the switching cycle lifetime: 1) Using an array of downscaled phase-change cells (instead of a continuous patch) each of which experiencing successful amorphization across its volume upon applying the reset pulse. Complete melting process at every reset switching and remixing of the elements fairly decrease the spatial segregation [*MRS bulletin* 39.8 (2014): 703-710]; 2) Leveraging interfacial phase-change structures where a superlattice configuration comprising of stacked Sb_2Te_3 and GeTe layers enables solid–solid (rather than typical solid-liquid-solid) phase transformation between the crystalline and amorphous states [*Nature nanotechnology* 6.8 (2011): 501-505]; 3) Using high-quality growth processes (such as molecular beam epitaxy and atomic layer deposition) to improve the interface qualities and reducing the grain sizes that facilitate uniform conversion across the material volume [*Materials Science in Semiconductor Processing* 137 (2022): 106244, *Journal of the American Chemical Society* 131.10 (2009): 3478-3480]; 4) Exploiting thermal layers such as graphene to reduce atomic migration during the phase transformation [*Science* 336.6088 (2012):

1561-1566]; 5) Discovering phase-change alloys showing zero-mass density change upon phase transition (such as GaSb and GeTe-CuTe) [*Materials Science and Technology* 33.16 (2017): 1890-1906]; 6) Minimizing the exposure of the PCM cell to the ambient by tight encapsulation of the PCM cell in a diffusion-free medium with high thermal conductivity. Since switching cyclability is also closely linked to the device design and the compounds surrounding the PCM, appropriate selection of materials and clever design of heterostructures can improve the number of switching cycles in phase-change metasurfaces. Despite the infancy of electrically reconfigurable phase-change nanophotonics, so far, a few thousand cycles of switching have been demonstrated in optical phase-change memories after which the device performance starts to degrade [*Advanced Materials* 32.31 (2020): 2001218]. We also expect a similar number of switching cycles with our platform though improvement in the cyclability is envisaged through collaborative efforts of material science and nanophotonics communities. **We have added Supplementary Note 3 in the revised version covering our discussion related to this comment.**

- Tuning of the P-GST states is characterized as quasi-continuous. What is the limit on the resolution of which states are achievable and how many states, in principal, could be accessed?

Our response: In principle, the effective dielectric constant of PCMs at different crystallization levels is modelled with an effective medium theory such as Lorentz-Lorenz relation. Continuous tuning of GST from amorphous to fully crystalline, through an infinite number of intermediate states characterized by unique optical constants, is offered by this model. However, in practice, such a luxury is found to be quasi-continuous, i.e., a limited number of intermediate states is accessible, depending on the dynamics of phase-change and properties of surrounding compounds. The crystallization mechanism of GST is defined as nucleation dominated [*Journal of Vacuum Science & Technology B, Nanotechnology and Microelectronics: Materials, Processing, Measurement, and Phenomena* 28.2 (2010): 223-262]; upon applying an external stimulus, nuclei are randomly formed in the amorphous matrix followed by their omni-directional growth due to the temperature rise until they impinge each other and form a uniform crystalline island in the bulk of the PCM. In addition to the stochastic nature of nuclei formation [*Nature nanotechnology* 11.8 (2016): 693-699], phase transformation is also influenced by the intrinsic properties of the substrate and capping layer materials as well as the roughness of embedding media [*Materials Science and Technology* 33.16 (2017): 1890-1906]. While in our current work, 4 distinguishable, reliable, and repeatable intermediate states are demonstrated through a single-pulse programming, up to 34 levels of crystallization levels has been recently reported in integrated photonic phase-change memories using a dual-pulse programming technique [*Optica* 6.1 (2019): 1-6]. To increase the intermediate levels, adoption of the following strategies for fine control over the fraction of crystalline nuclei is imperative. First and foremost is employing of an iterative signal programming scheme in which: i) a sequence of gradually decreasing power set pulses, or ii) a packet of consecutive pulses with the same power is followed by sequential verification steps, that is the reflectance measurement of the metasurface [*Nature photonics* 9.11 (2015): 725-732, *IEEE International Symposium of Circuits and Systems (ISCAS)*. *IEEE* (2011), 329-332]. Each of these pulses initiates a partial crystallization in a feedback loop leading to precise control over the crystallographic state of GST. In addition to the iterative programming, leveraging a pulse shaping technique with controllable amplitude and width of the electrical pulse(s) can be beneficial for the

multi-level operation of the phase-change cell [*Optica* 6.1 (2019): 1-6]. Along with such coding and signal processing strategies, other plans such as i) leveraging an array of miniaturized cells formed by engineered heterostructures that comprise ultrathin layers of the PCM, ii) synthesizing new PCMs characterized by minimized change in configurational entropy between amorphous and crystalline states, and iii) investigation of compositionally tuned PCMs with engineered nucleation and growth rates that relax the stochastic behavior of GST can be pursued in future efforts. **We have clarified the above point by adding Note 5 in Supplementary Information.**

- Is there any indication as to the nature of the t-SNE dimensions of the latent space? It is evident that A-GST and C-GST structures access different regions within this space, but it is not apparent how to interpret a given point and relate it to the physical response of the metasurface. Particularly in regards to the statement: "...incorporation of GST in the meta-atom considerably spans the attainable responses not accessible through just variation of structural parameters with one GST state.", how are the responses attainable only with C-GST meaningfully different from the responses only achievable with A-GST? It seems premature to conclude that the crystallization state is an ideal tuning knob, if it's not understood what exactly is being tuned (i.e., what it means to travel from one point to another within the latent space).

Our response: We would like to emphasize that t-SNE as a state-of-the-art non-parametric, nonlinear dimensionality-reduction technique with major purpose of data visualization preserves the pairwise distances between data points [*JMLR* 9.86, 2579-2605 (2008), *Data Mining and Knowledge Discovery* 5.2 (2015): 51-73]. The essence of using t-SNE is to trustfully represent complex data sets with intrinsically high dimensions in low-dimensional spaces while preserving as much relevant information as possible [*Neural Computation* 24.3 (2012): 771-804, *Distill* 1.10 (2016): e2]. However, t-SNE can uncover hidden structures in the high-dimensional dataset through capturing much of the local structures, while also revealing global structure such as the presence of clusters [*Journal of machine learning research* 9.11 (2008): 2579-2605, *Nature biotechnology* 37.1 (2019): 38-44]. This is possible due to two unique features of t-SNE. First, this algorithm cares about the pairwise distances between data points, which is realized by translating Euclidean distances between data points in the high-dimensional space into conditional probabilities that represent "similarities" [*Journal of Machine Learning Research* 15.1 (2014): 3221-3245]. In projection of the high-dimensional data to the low-dimensional space, t-SNE tends to position the points on a plane (or hyperplane) such that the pairwise distances minimize a cost function, which is a measure of the similarity between two probability distributions [*Distill* 1.10 (2016): e2]. In addition, t-SNE fairly ameliorate the "crowding problem" distinguished by the overlap of projected datapoints in the center of the low-dimensional map due to the excessive attractive forces between moderately distant datapoint in the original space [*Journal of machine learning research* 9.11 (2008): 2579-2605]. In this regard, distinctly isolated clusters of similar datapoints sharing multiple features can be readily identified in the low-dimensional space, though t-SNE is not a dedicated clustering algorithm [*Nature methods* 16.3 (2019): 243-245, *Nature communications* 10.1 (2019): 1-14, *SIAM Journal on Mathematics of Data Science* 1.2 (2019): 313-332]. In our case, the algorithm forms two widely separated natural clusters of similar points, i.e., for A-GST and C-GST cases (blue and red dots in Fig. 4a). The minimum overlap between the two clusters implies that despite the variation of the structural parameters, each state of GST

can only provide a specific cluster of datapoints, which are different in nature from the other cluster. We think our wording in the original manuscript did not convey the main message, which is the emphasis on “two widely unfolded clusters” and not “span of the full 3D t-SNE latent space”. Noteworthy, the axes of a t-SNE plot are abstract scores describing complex curved paths in the original space and are not meant to be straightforwardly interpretable in terms of the axis/units of the original high-dimensional space. This is the reason why t-SNE plots without scaling are also common in the literature. Unlike simple linear dimensionality reduction algorithms such as principal component analysis (PCA) whose plot axes are weighted linear combinations of the original dimensions, as specified by the principal eigenvectors of the covariance matrix, the axes in t-SNE plots do not convey any specific information [*Distill 1.10 (2016): e2*]. **Having said that, to avoid any misinterpretation, we have rephrased the corresponding paragraph in the revised manuscript as follows: “The implication of the formation of two distinguishable unfolded clusters corresponding to A-GST (blue points) and C-GST (red points) in the 3D latent space is twofold. Incorporation of GST in the meta-atom grants a different class of responses not easily accessible through just variation of structural parameters with one GST state. Accordingly, the GST crystallization state can be considered as an effective tuning knob to modify the metasurface performance. Furthermore, the metasurfaces with A-GST and C-GST are likely governed by modes with distinct natures.”.**

We would like to conclude that while visual representation of high-dimensional data in a low-dimensional space is imperative for exploratory data analysis, certain attention should be paid to prevent some common misreading of t-SNE plots. 1) The relative distance between well-separated clusters in the low-dimensional space does not provide any intuition. In fact, t-SNE does not retain distances but probabilities, so Euclidean distances in high-dimensional and low-dimensional spaces do not provide any useful measure about the response of the metasurface. 2) The relative sizes of the clusters in t-SNE plots do not convey any specific meaning. This stems from the fact that t-SNE tends to expand/contracts the dense/sparse data clusters of the high-dimensional space in the projected embeddings. 3) The topology of generated clusters in the latent space does not provide any information about the nature of the dataset in the original dimension. The algorithm hyperparameters fairly affect the elongation, curvature, or clumping of the generated clusters. 4) Successive runs of the t-SNE algorithm do not necessarily generate similar outputs. This follows from the rationale that some hyperparameters like perplexity, a measure of the effective nearest neighbors, play a key role in the optimization process. Despite its multiple strengths in data visualization, t-SNE has some weaknesses. The main functionality of t-SNE is for visualization purposes so using it as a pre-processing machine-learning approach is tricky, if not improper. Also, t-SNE has a low performance on datasets with high intrinsic dimensions especially given the nonconvexity of the cost function that requires optimization of several hyperparameters.

To complete our discussion, we investigate the evolution of electric field distribution for three simulated metasurfaces with different structural parameters for both A-GST and C-GST cases (see Supplementary Fig. S11a, also shown below). For each state of GST, comparable mode profiles can be observed with all 3 randomly selected samples. While the top row illustrates the excitation of short-range surface plasmon mode at the interface of the gold nanodisk and the alumina layer, the field enhancement in the bottom row mainly occurs at the tips of the gold nanodisk and interface of the gold back-reflector due to the excitation of long-range surface plasmon polaritons.

The reduced-dimensional responses are indicated in the 3D latent space in Supplementary Fig. S11b (shown below) using color-coded shapes. While this should not be interpreted as the “physics extraction capability”, at least it gives a tangible insight on the linking between some randomly selected datapoints within one cluster to the mode profile of the corresponding meta-atoms. **We hope the newly added discussions in both the revised manuscript, Supplementary Note 6, and Supplementary Fig. S11 will help the interpretations of the t-SNE analysis.**

Figure R10: (a) Inspection of the normalized electric field magnitude at the resonance wavelengths of the meta-switch with A-GST (top row) and C-GST (bottom row) in the $x-z$ plane of a meta-atom. The structural parameters of the studied metasurface are $p = 440$ nm, $d_{Au} = 210$ nm, and $t_{GST} = 25$ nm (first column), $p = 580$ nm, $d_{Au} = 190$ nm, $t_{GST} = 35$ nm (second column), and $p = 750$ nm, $d_{Au} = 230$ nm, $t_{GST} = 45$ nm (third column). (b) The embeddings corresponding to A-GST and C-GST in the latent space. The reduced-dimensional reflectance responses of metasurfaces in panel (a) are depicted using color-coded shapes.

REVIEWERS' COMMENTS

Reviewer #1 (Remarks to the Author):

I am satisfied with the comprehensive/clear responses from authors, I believe the paper is ready for publication.

Reviewer #2 (Remarks to the Author):

Abdollahramezani and his co-workers have made a remarkable effort in improving the manuscript in a significant way. By adding a new experimental case study (tunable beam deflector) to the revision, the authors have totally eradicated my concerns about missing novelty. On the contrary, I want to congratulate the authors for having added these nice, novel results which are furthermore illustrated in an excellent way. Even though no full 2π are achieved yet, the results are truly impressive. Pointing out the novelty aspects of the phase-change platform itself in the introductions also helps the reader to put the present work into context.

I personally want to thank the authors for their extensive responses also to the questions that I posed more out of personal curiosity and which were less helping to improve the manuscript.

Finally, I am still not convinced that the t-SNE dimensionality reduction section in the main text is of great use, and even the authors actually discuss the dangers and difficulties of the interpretability themselves in their response letter.

But at least the authors have removed the ambiguous discussion from the manuscript, that raised the expectation that the analysis revealed insights in the physical mechanisms, or that the full span of latent space by data points has any meaning in the interpretation.

In summary, thanks to the extensive and careful revision and thanks to the detailed reply to all open questions, I am glad to now fully recommend the manuscript for publication with no further revisions.

Reviewer #3 (Remarks to the Author):

Thank you for clarifying about the raised concerns. I believe the manuscript is suitable for publication.

Reviewer #1 (Remarks to the Author):

I am satisfied with the comprehensive/clear responses from authors, I believe the paper is ready for publication.

Our response: We appreciate the Reviewer for the positive comment and support of publication in Nature Communications.

Reviewer #2 (Remarks to the Author):

Abdollahramezani and his co-workers have made a remarkable effort in improving the manuscript in a significant way. By adding a new experimental case study (tunable beam deflector) to the revision, the authors have totally eradicated my concerns about missing novelty. On the contrary, I want to congratulate the authors for having added these nice, novel results which are furthermore illustrated in an excellent way. Even though no full 2π are achieved yet, the results are truly impressive. Pointing out the novelty aspects of the phase-change platform itself in the introductions also helps the reader to put the present work into context.

I personally want to thank the authors for their extensive responses also to the questions that I posed more out of personal curiosity and which were less helping to improve the manuscript.

Finally, I am still not convinced that the t-SNE dimensionality reduction section in the main text is of great use, and even the authors actually discuss the dangers and difficulties of the interpretability themselves in their response letter. But at least the authors have removed the ambiguous discussion from the manuscript, that raised the expectation that the analysis revealed insights in the physical mechanisms, or that the full span of latent space by data points has any meaning in the interpretation.

In summary, thanks to the extensive and careful revision and thanks to the detailed reply to all open questions, I am glad to now fully recommend the manuscript for publication with no further revisions.

Our response: We appreciate the Reviewer's interest in our revised manuscript and clear support of our work for publication. We also thank the Reviewer for the positive comment on the added novelty and for the positive comment about our responses to the open questions.

Reviewer #3 (Remarks to the Author):

Thank you for clarifying about the raised concerns. I believe the manuscript is suitable for publication.

Our response: We thank the Reviewer for considering our revised manuscript and recommending the work for publication.